# Set-Preserving Calibration from Conformal P-Values to E-Values

**Nabil Alami** [1]   **Jad Zakharia** [2]   **Souhaib Ben Taieb** [1 3]

## Abstract

Standard conformal prediction (CP) procedures are typically formulated in terms of p-values, but reliance on p-values alone limits flexibility, for example, when combining dependent evidence across models or data splits. Recent work has explored e-value formulations for conformal inference, yet a direct connection between p- and e-value formulations in CP has been missing, especially regarding their statistical efficiency. We first identify limitations of classical p-to-e calibrators in the CP setting, showing that they are not set-preserving and can lead to overly conservative prediction sets. To address this, we propose a novel P2E calibrator that converts conformal p-values into e-values without altering the prediction set induced by the original conformal p-value. We establish both theoretically and empirically that our calibrator can yield significant efficiency gains over existing p-to-e calibrators. This e-value formulation enables principled use of recent advances in e-value merging and randomization, where we demonstrate its impact in two applications: cross-conformal prediction (CCP), whose variants typically provide only approximate $1 - 2\alpha$ coverage, and conformal aggregation (CA). In both cases, our e-value-based methods satisfy the desired $1 - \alpha$ coverage guarantee while improving efficiency over standard baselines. More broadly, our approach expands the flexibility of CP and opens new directions for efficient, distribution-free uncertainty quantification.

Modern machine learning models are increasingly deployed in high-stakes domains where uncertainty is as critical as accuracy. Conformal prediction (CP) has emerged as a versatile framework for constructing prediction sets in both regression and classification tasks (Shafer & Vovk, 2008; Angelopoulos et al., 2023; Papadopoulos et al., 2002; Vovk et al., 2005). Given any black-box predictor, conformal methods output a prediction set for a new test input that guarantees finite-sample marginal coverage: with user-specified probability (e.g., 90%), the set contains the true outcome.

For decades, CP has been grounded in p-values (Vovk, 2025). Classical p-value–based approaches are often efficient in practice, yielding tight prediction sets while maintaining finite-sample coverage. However, their theoretical flexibility is limited, particularly when aggregating dependent p-values (Vovk et al., 2022). In contrast, recent work on e-value–based CP (Balinsky & Balinsky, 2024; Gauthier et al., 2025; Koning & van Meer, 2025) demonstrates how e-values extend conformal methods to settings less suited for p-values. Originally introduced to the statistical community by Vovk & Wang (2021), *e-values* offer more flexible aggregation rules and additional advantages for hypothesis testing (Vovk & Wang, 2021; 2024; Wang, 2025). Nevertheless, it remains unclear whether e-value–based conformal methods suffer efficiency losses compared to their p-value counterparts, and no unifying principle seems to have been established to connect p-values and e-values within CP.

To address this gap, we propose a novel framework that links conformal p-values and e-values through a *P2E calibrator*, after exposing some of the limitations of classic p-to-e calibrators (Vovk & Wang, 2021; Ramdas & Wang, 2025). We show how our P2E calibrator is *set-preserving*, which ensures that the e-value formulation of CP is decision-equivalent to the standard p-value formulation: for any test point and miscoverage level, the resulting prediction set is identical. In doing so, our method can yield tight prediction sets while opening the door to the broader e-value theory, including useful extensions based on Markov-type inequalities and randomization results (Ramdas & Manole, 2026).

Our framework thus provides a principled way to translate CP from a p-value formulation into an e-value formulation, combining the strengths of both approaches. In our view, p-values and e-values are not competitors but complementary tools that together enrich conformal prediction and its applications.

In summary, our contributions are as follows:

[1]Department of Statistics and Data Science, MBZUAI [2]École Normale Superieure de Cachan [3]Department of Computer Science, University of Mons. Correspondence to: Nabil Alami <nabil.alami@mbzuai.ac.ae>.

*Proceedings of the 43rd International Conference on Machine Learning*, Seoul, South Korea. PMLR 306, 2026. Copyright 2026 by the author(s).

1. We study the integration of classic p-to-e calibrators in CP while highlighting their limitations.

2. We introduce a novel P2E calibrator which transforms the conformal p-variable into an e-variable which preserves the same CP set, and offers additional advantages over classic p-to-e calibrators.

3. We leverage e-value theory and Markov-type inequalities combined with our P2E calibrator framework to enhance two settings: cross-conformal prediction (CCP) and conformal aggregation (CA).

4. We show experimentally that our e-value–based CP methods improve efficiency and flexibility compared to their p-value alternatives.

# 1. Background and notations

We consider a regression problem where the goal is to predict a real-valued response $y \in \mathcal{Y} = \mathbb{R}$ from an input vector $x \in \mathcal{X} \subseteq \mathbb{R}^p$. We are given a dataset $\mathcal{D} = \{(X_i, Y_i)\}_{i \in \mathcal{I}}$, where the pairs $(X_i, Y_i)$ are exchangeable samples from an unknown joint distribution $Q_{X,Y}$ defined over $\mathcal{X} \times \mathcal{Y}$.

Let $\hat{\mu} : \mathcal{X} \to \mathcal{Y}$ denote a base regressor trained on $\mathcal{D}$. When the model is trained on a subset $\mathcal{D}_1 \subset \mathcal{D}$, we write $\hat{\mu}(\mathcal{D}_1)$ to explicitly indicate the dependence on the training data. Given a miscoverage $\alpha \in (0, 1)$, CP allows to construct for any new input $X_{n+1}$ of unknown response $Y_{n+1}$ a prediction set $\mathcal{C}(X_{n+1}) \subseteq \mathcal{Y}$ that satisfies the finite-sample coverage guarantee (Vovk et al., 2005).

## 1.1. Conformal Prediction with p-values

Split-conformal prediction (SCP) (Papadopoulos et al., 2002) is an efficient variant of CP that partitions the dataset $\mathcal{D}$ into a training set $\mathcal{D}_{\text{train}}$ and a calibration set $\mathcal{D}_{\text{cal}}$. Let $\mathcal{I}$ be the index set of $\mathcal{D}$, split as $\mathcal{I} = \mathcal{I}_{\text{train}} \sqcup \mathcal{I}_{\text{cal}}$, where $\mathcal{I}_{\text{cal}} := \{1, \ldots, n\}$.

The base model is first trained on $\mathcal{D}_{\text{train}}$ to obtain a predictor $\hat{\mu}$. Then we define a nonconformity score function $s_{\hat{\mu}} : \mathcal{X} \times \mathcal{Y} \to \mathbb{R}$ (e.g. $s_{\hat{\mu}}(x, y) = |y - \hat{\mu}(x)|$ in regression) and compute the nonconformity scores on the calibration set $\mathcal{S}_{\text{cal}} = \{s_i := s_{\hat{\mu}(\mathcal{D}_{\text{train}})}(X_i, Y_i) : i = 1, \ldots, n\}$.

The SCP prediction set is

$$\mathcal{C}(X_{n+1}) = \{ y \in \mathcal{Y} : s_{\hat{\mu}(\mathcal{D}_{\text{train}})}(X_{n+1}, y) \leq s_{(k)} \}, \quad (1)$$

where $s_{(\ell)}$ denotes the $\ell$-th order statistic of $\mathcal{S}_{\text{cal}}$ and $k = \lceil (1 - \alpha)(n + 1) \rceil$.

Under the assumption that $\mathcal{D}_{\text{cal}} \cup (X_{n+1}, Y_{n+1})$ are exchangeable, the prediction set (1) satisfies

$$\mathbb{P}(Y_{n+1} \in \mathcal{C}(X_{n+1})) \geq 1 - \alpha. \quad (2)$$

The definition of the prediction set in (1) can be equivalently expressed in terms of p-values. Instead of comparing the test score directly to the empirical quantile $s_{(k)}$, one can compute the *conformal p-value* for any candidate label $y \in \mathcal{Y}$ as

$$P_n(y) = \frac{1 + \sum_{i=1}^n \mathbf{1}(s_i \geq s_{\hat{\mu}(\mathcal{D}_{\text{train}})}(X_{n+1}, y))}{n + 1}, \quad (3)$$

The corresponding prediction set is

$$\mathcal{C}^{\mathbf{pv}}(X_{n+1}) = \{ y \in \mathcal{Y} : P_n(y) > \alpha \}. \quad (4)$$

In what follows, we assume no ties among nonconformity scores (which is natural in the context of regression). Then, under exchangeability $P_n(y)$ is uniformly distributed on $\{1/(n+1), \ldots, 1\}$ under the null hypothesis $\mathbb{H}_0 : Y_{n+1} = y$ (Vovk et al., 2005; Lei et al., 2018). Hence $P_n$ is a valid p-variable, i.e.,

$$\mathbb{P}_{\mathbb{H}_0}(P_n(y) \leq \delta) \leq \delta, \quad \forall \delta \in (0, 1).$$

This formulation shows that the CP set consists exactly of the candidates $y$ for which the conformal p-value exceeds $\alpha$. We will refer to this p-value formulation as p-CP.

## 1.2. Conformal Prediction with e-values

Similarly to p-variables, *e-variables* can be used to test the null hypothesis $\mathbb{H}_0$. E-variables were introduced by Vovk & Wang (2021) and offer several advantages over p-variables (Vovk & Wang, 2021; 2024; Wang, 2025). The realizations of e-variables are referred to as *e-values*.

**Definition 1.1.** (Vovk & Wang, 2021; Ramdas & Wang, 2025) An *e-variable* $E$ for a null hypothesis $\mathbb{H}$ is a random variable taking values in $[0, +\infty)$ satisfying $\mathbb{E}_{\mathbb{H}}[E] \leq 1$. It is said to be *exact* if $\mathbb{E}_{\mathbb{H}}[E] = 1$.

Given an e-variable $E$ with realized e-value $e$, a level-$\alpha$ test rejects $\mathbb{H}_0$ whenever $e \geq 1/\alpha$. By Markov's inequality (MI), we obtain the following Proposition.

**Proposition 1.2.** *Let $E$ be an e-variable and $\alpha \in (0, 1)$. Then $\mathbb{P}(E < 1/\alpha) \geq 1 - \frac{\mathbb{E}_{\mathbb{H}}[E]}{1/\alpha} \geq 1 - \alpha$.*

Therefore, replacing the P-variable $P_n$ in (3) with an E-variable $E_n$ as the test statistic yields the conformal prediction set

$$\mathcal{C}^{\mathbf{ev}}(X_{n+1}) = \{ y \in \mathcal{Y} : E_n(y) < 1/\alpha \},$$

which has valid finite-sample coverage as a direct consequence of Proposition 1.2. We will refer to this e-value formulation as e-CP.

Balinsky & Balinsky (2024) proposed such e-variable of the form

$$E_n(y) := \frac{(n + 1)\, s_{\hat{\mu}}(X_{n+1}, y)}{\sum_{i=1}^n s_i + s_{\hat{\mu}}(X_{n+1}, y)}, \quad y \in \mathcal{Y}, \quad (5)$$

which is exact, and can be directly employed within the CP framework.

One advantage of e-CP is that it naturally connects CP to the broader theory of e-values, thereby inheriting many of their desirable properties. Recent work by Gauthier et al. (2025; 2026) demonstrates how e-CP generalizes classical CP in certain applications. However, an important open question remains: *how efficient is e-CP compared to p-CP?*

For the remainder of the paper, we omit the explicit notation '$y \in \mathcal{Y}$' when writing prediction sets.

### 1.3. Fundamental properties of e-values

We briefly review several properties of e-values, mainly derived from Vovk & Wang (2021). These properties will later be leveraged in our e-CP methods.

**E-Merging.** Merging e-variables is done via *e-merging functions*, which essentially transform a vector of $K \geq 1$ e-variables $\mathbf{E} = (E_1, \ldots, E_K)$, for any hypothesis, into a single e-variable $M(\mathbf{E})$. For example, it is easy to check that the arithmetic mean is an e-merging function. An e-merging function $M$ is said to be *admissible*, if there is no other e-merging function $N$ such that $N \geq M$ (pointwise) and $N(\mathbf{e}) > M(\mathbf{e})$ for some $\mathbf{e}$. Unlike p-values, e-values admit simple admissible merging rules that remain valid even under arbitrary dependence.[1]

**Proposition 1.3.** *The arithmetic mean essentially dominates any symmetric e-merging function.*

**Proposition 1.4** (Wang (2025)). *The only admissible merging functions for arbitrary e-values are weighted arithmetic means.*

Given a vector of $K$ e-variables $\mathbf{E} = (E_1, \ldots, E_K)$, the choice of e-merging function directly affects the efficiency of the resulting e-CP set. Indeed, if $M_1 \geq M_2$ are two e-merging functions, then for any $\alpha \in (0, 1)$ we have $\{M_1(\mathbf{E}) < 1/\alpha\} \subseteq \{M_2(\mathbf{E}) < 1/\alpha\}$, while both sets satisfy coverage.

**Extensions of MIs.** The practicality of e-values stems in part from their connection to Markov's Inequality, which enables direct thresholding. We present here two useful extensions of MI when working with e-variables.

**Proposition 1.5** (Ramdas & Manole (2026)). *Let $E$ be an e-variable, independent of $U \sim \mathrm{Unif}(0,1)$. Then,*

$$\mathbb{P}\left(E > \tfrac{U}{\alpha}\right) \leq \alpha,$$

---

[1] By contrast, the arithmetic mean of $K$ p-values $p_1, \ldots, p_K$ is generally not a p-value; at best one has $\mathbb{P}\left(\frac{1}{K}\sum_{i=1}^{K} p_i \leq \alpha\right) \leq 2\alpha$, and the constant 2 is sharp.

*which is referred to as the* Uniformly Randomized Markov's Inequality (UR-MI).

**Proposition 1.6** (Manole & Ramdas (2023)). *Let $E_1, \ldots, E_n$ be exchangeable e-variables, and let $U \sim \mathrm{Unif}(0,1)$ be independent. Then,*

$$\mathbb{P}\left(\left\{E_1 \geq \tfrac{U}{\alpha}\right\} \cup \left\{\sup_{k \leq n} \frac{1}{k}\sum_{i=1}^{k} E_i \geq \tfrac{1}{\alpha}\right\}\right) \leq \alpha.$$

## 2. A new set-preserving P2E calibrator

This section establishes the core technical foundation of the paper: a novel calibrator with useful properties from the conformal p-variable to an e-variable yielding tight prediction sets. We focus on CP problems involving the aggregation of multiple p-values.

E-variables can be constructed in different ways, but it is generally unclear how a given design choice affects the constructed prediction set. For example, we show in Appendix A that the construction in (5) can lead to overly conservative prediction sets in the SCP framework. We therefore start from the conformal p-variable and calibrate it into an e-variable, building on the existing calibration literature.

The proofs of subsequent results are found in Appendix B.

### 2.1. Limitations of classical P-to-E calibrators

**P-to-e calibrators.** A standard tool for converting p-values into e-values is a *p-to-e calibrator* (Vovk & Wang, 2021; Ramdas & Wang, 2025).

**Definition 2.1.** A *p-to-e calibrator* is a decreasing function $F : [0, 1] \to [0, \infty]$ such that, for any hypothesis $\mathcal{H}$ and any p-variable $P$ for $\mathbb{H}$, $F(P)$ is an e-variable for $\mathbb{H}$.

Examples of p-to-e calibrators include $F_1(p) := -\log(p)$, $F_2(p) := p^{-1/2} - 1$, and $F_3(p) := 2(1 - p)$. We also highlight the *all-or-nothing* (AoN) calibrator (for a fixed $\alpha \in (0, 1)$):

$$F_{\mathrm{AoN}}(p) := \frac{1}{\alpha}\mathbf{1}_{\{p \leq \alpha\}}.$$

Thus, given the conformal p-variable $P_n$ in (3), $F_1(P_n), F_2(P_n), F_3(P_n)$ and $F_{AoN}(P_n)$ are all e-variables for $\mathbb{H}_0$. We can then use these e-variables for conformal inference, from which the $1 - \alpha$ coverage guarantee is recovered simply by thresholding at $1/\alpha$ as explained in Section 1.2.

**Set-Preservation and Efficiency.** Although coverage is immediate once we have an e-variable, the choice of the p-to-e calibrator $F$ can substantially impact the size of the resulting prediction sets. Since predicting this efficiency

effect from the calibrator alone is difficult, as it also depends on the underlying score distribution, and the specific construction of the prediction set, we focus on a structural property of the calibrator: *set-preservation*.

**Definition 2.2.** A p-to-e calibrator $F$ is said to be *set-preserving* (at level $\alpha$) if, for all $n$ and for any nonconformity score used to build the p-variable $P_n$ in (3), we have

$$\mathcal{C}^{\mathbf{pv}}(X_{n+1}) = \{P_n > \alpha\} = \{E_n < 1/\alpha\} = \mathcal{C}^{\mathbf{ev}}(X_{n+1}).$$

where $E_n := F(P_n)$. It is said to be *set-inflating* if

$$|\{P_n > \alpha\}| \leq |\{E_n < 1/\alpha\}|.$$

Set-preservation ensures that the conformal p-variable and its calibrated e-variable induce the same SCP set, regardless of the calibration size and nonconformity score. Set-inflation means that the calibrated e-value set has size at least as large as the standard SCP set, and can therefore be viewed as an efficiency criterion. We characterize the class of calibrators (independent of the calibration set size $n$) satisfying this property:

**Proposition 2.3.** *Among all left-continuous p-to-e calibrators, only $F_{\mathrm{AoN}}$ is set-preserving.*

To provide intuition and illustrate the significance of set-preservation, consider an aggregation context where we have $K$ conformal p-variables $\{P_n^{(k)}\}_{k=1}^K$ (e.g., from $K$ predictors, or $K$ folds) for the same hypothesis $\mathbb{H}_0$, and their calibrated e-variables $E_n^{(k)} := F(P_n^{(k)}), k = 1, \ldots, K$. A natural e-merging rule is to average the e-variables, yielding the prediction set:

$$\mathcal{C}^F_{\mathbf{Agg}}(X_{n+1}) := \Big\{ \frac{1}{K} \sum_{k=1}^K E_n^{(k)} < \frac{1}{\alpha} \Big\}. \qquad (6)$$

We denote the set $\mathcal{C}^F_{\mathbf{Agg}}(X_{n+1})$ to emphasize the dependence on $F$. Through union and intersection bounds, we obtain:

$$\bigcap_{k=1}^K \{E_n^{(k)} < 1/\alpha\} \subseteq \mathcal{C}^F_{\mathbf{Agg}}(X_{n+1}) \subseteq \bigcup_{k=1}^K \{E_n^{(k)} < 1/\alpha\}.$$

When the calibrator $F$ is not set-preserving, the individual e-CP sets $\mathcal{C}^{\mathbf{ev}}_k(X_{n+1}) := \{E_n^{(k)} < 1/\alpha\}$ are larger than the SCP sets $\mathcal{C}^{\mathbf{pv}}_k(X_{n+1}) := \{P_n^{(k)} > \alpha\}$. This expansion widens the resulting sandwich bound, and explains why it may lead to overly conservative prediction sets. Conversely, if $F$ is set-preserving, the following sandwich bound holds:

$$\bigcap_{k=1}^K \mathcal{C}^{\mathbf{pv}}_k(X_{n+1}) \subseteq \mathcal{C}^F_{\mathbf{Agg}}(X_{n+1}) \subseteq \bigcup_{k=1}^K \mathcal{C}^{\mathbf{pv}}_k(X_{n+1}),$$

This justification gives intuition on why set-preserving calibrators can yield tight prediction sets. This behavior is

further confirmed empirically in Section 5, where we observe that set-inflating p-to-e calibrators (like $F_1, F_2, F_3$) result in significantly larger sets.

Consequently, to maintain statistical efficiency, Proposition 2.3 indicates that the only p-to-e calibrator one should use is $F_{\mathrm{AoN}}$. However, the latter has some limitations as it can take zero values which destroys statistical evidence. Indeed, this can create a practical and theoretical drawback in sequential and optional-stopping contexts, where products of e-values $\prod_t E_t$ naturally arise: a single zero e-value collapses the entire product and prevents accumulation of evidence. A similar issue can also arise in aggregation: under independence, the product is a well defined e-merging function (Vovk & Wang, 2021), but again a single zero forces the merged e-value to zero[2]. Since our framework can also support these broader uses of e-values, we seek an alternative.

## 2.2. Our P2E calibrator

We propose a novel set-preserving **P2E calibrator**, which depends on both $n$ and $\alpha$, denoted by $F_{n,\alpha}$, that addresses the limitations discussed above. Specifically, it satisfies three conditions:

**A.** $F_{n,\alpha}$ is set-preserving,

**B.** $E_{n,\alpha} := F_{n,\alpha}(P_n)$ is an exact e-variable[3],

**C.** $F_{n,\alpha}$ is smooth, invertible, and strictly positive.

Let $\mathcal{E}$ denote the class of all positive, strictly decreasing real functions on $[0, 1]$. For any $f \in \mathcal{E}$, the p-CP set in (4) can equivalently be written as

$$\mathcal{C}^{\mathbf{pv}}(X_{n+1}) = \{f(P_n) < f(\alpha)\} = \Big\{ \frac{f(P_n)}{\alpha f(\alpha)} < 1/\alpha \Big\}. \qquad (7)$$

Our key idea is to choose a suitable function $f \in \mathcal{E}$ such that the ratio $f(P_n)/\alpha f(\alpha)$ is itself an e-variable. The following proposition asserts the existence of such a function.

**Proposition 2.4.** *Let $n \geq 1$, and let $P_n$ be the p-variable in (3). For $\alpha \in (0, 1)$, suppose $\alpha(n + 1) \in (1, \infty) \setminus \mathbb{N}$. Then, there exists a function $f_{n,\alpha} \in \mathcal{E}$ such that*

$$\mathbb{E}_{\mathbb{H}_0}\Big( \frac{f_{n,\alpha}(P_n)}{\alpha f_{n,\alpha}(\alpha)} \Big) = 1. \qquad (8)$$

We denote by $\mathcal{E}_{n,\alpha}$ the set of all functions of the form $p \to f_{n,\alpha}/\alpha f_{n,\alpha}(\alpha)$ of Proposition 2.4, that we call **P2E**

---

[2]this relates to the notion of e-power (Ramdas & Wang, 2025), defined as $\mathbb{E}_{\mathbb{H}}[\log E]$. The e-power of $F_{AoN}$ is equal to $-\infty$.

[3]Exactness avoids unnecessary conservativeness: if $\mathbb{E}[E] = c < 1$, then the exact e-variable $E/c$ yields the smaller set $\{E/c < 1/\alpha\} \subseteq \{E < 1/\alpha\}$, with both satisfying coverage by MI.

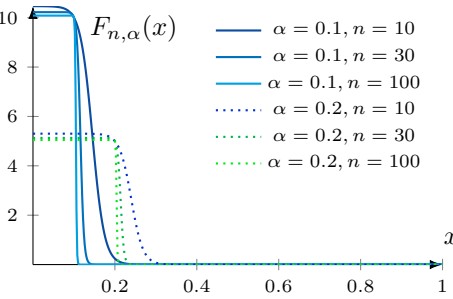

*Figure 1.* Behavior of the mapping $x \mapsto F_{n,\alpha}(x)$ for varying values of $n$ and $\alpha$.

**calibrators**. Recall that these functions differ from standard p-to-e calibrators, as they depend on both $\alpha$ and $n$. Additionally, we show that all P2E calibrators necessarily converge to $F_{AoN}$.

**Proposition 2.5.** *For any sequence $F_n \in \mathcal{E}_{n,\alpha}$, we have*

$$F_n(p) \xrightarrow{n \to \infty} F_{\text{AoN}}(p)$$

*for every $p \in (0,1)$.*

Thus, Proposition 2.5 establishes a performance guarantee for P2E calibrators: *Unlike standard p-to-e calibrators, they are set-preserving and asymptotically recover the behavior of $F_{AoN}$, which is itself set-preserving.*

We now present our main theorem, providing an explicit P2E calibrator which satisfies the three conditions **A-C**.

**Theorem 2.6.** *Let $P_n$ be the p-variable in* (3) *and suppose $\alpha(n+1) \in (1,\infty) \setminus \mathbb{N}$. Then, for any $s \in \left( \alpha, \frac{\lceil \alpha(n+1) \rceil}{n+1} \right)$, there exists $C_{n,\alpha} > 0$ such that*

$$F_{n,\alpha}(p) := \frac{1}{\alpha} \cdot \frac{1 + \exp\big(C_{n,\alpha}(\alpha - s)\big)}{1 + \exp\big(C_{n,\alpha}(p - s)\big)} \qquad (9)$$

*is in the set $\mathcal{E}_{n,\alpha}$. Additionally, $F_{n,\alpha} \geq F_{AoN}$ (pointwise).*

We visualize in Figure 1 the graph of $F_{n,\alpha}$ of Theorem 2.6.

The last domination property of the theorem is very relevant in our context. Indeed, when aggregating $K$ p-variables $P_n^{(1)}, ..., P_n^{(K)}$, it holds that $\frac{1}{K}\sum_{k=1}^{K} F_{n,\alpha}(P_n^{(k)}) \geq \frac{1}{K}\sum_{k=1}^{K} F_{\text{AoN}}(P_n^{(k)})$, which implies, following the notation of (6), that

$$\mathcal{C}_{\text{agg}}^{F_{n,\alpha}}(X_{n+1}) \subseteq \mathcal{C}_{\text{agg}}^{F_{\text{AoN}}}(X_{n+1}). \qquad (10)$$

Thus, our P2E calibrator is always more efficient than $F_{AoN}$ in this aggregation setting.

For the remainder of this paper, we adopt the P2E calibrator of Theorem 2.6.

## 3. Related Work

CP (Saunders et al., 1999; Vovk et al., 2005) is a distribution-free framework for uncertainty quantification; see Fontana et al. (2023); Angelopoulos et al. (2023) for surveys. SCP (Papadopoulos et al., 2002; Lei et al., 2018) requires only one model fit but can be inefficient, while Full-CP (Vovk et al., 2005) is more efficient but computationally costly. Intermediate approaches such as CCP (Vovk, 2015; Vovk et al., 2018), multi-split (Solari & Djordjilović, 2022), jackknife+ (Barber et al., 2021), and out-of-bag conformal methods (Gupta et al., 2022) aim to balance efficiency and computation by reusing data more effectively. CP has also been applied to aggregation, where the aim is to combine model uncertainties into a single prediction set. Some methods aggregate the prediction sets (Yang & Kuchibhotla, 2025; Gasparin & Ramdas, 2024), while others operate on the nonconformity scores (Luo & Zhou, 2025; Patel et al., 2026; Alami et al., 2026).

CP with e-values is discussed in Vovk (2025), where it is shown to be conceptually simpler than p-CP. A key advantage is that e-values, which can be transformed into p-values and vice-versa, offer many beneficial properties for hypothesis testing (Ramdas & Wang, 2025; Vovk & Wang, 2021; 2024; Wang, 2025).

In the CP framework, (Balinsky & Balinsky, 2024) independently design an e-variable based on exchangeable scores for traditional CP, while Koning (2025) derive a related more general class of e-values valid for conformal inference. Building on (Balinsky & Balinsky, 2024), Gauthier et al. (2025; 2026) extend CP to regimes where p-values do not fit; however, their methods can lack efficiency, which we believe is due to the e-variable design. Additionally, none of these works make an explicit link with classical p-CP. In our context, we study the link between e-CP and p-CP through p-to-e calibrators. Perhaps the closest approach to P2E is (Bickel, 2025), which gives sample-size-dependent mappings from p-values to Bayes factors and not in the CP context.

## 4. Our P2E calibrator For Cross-Conformal Prediction and Aggregation

**E-Cross-Conformal Prediction.** Cross-Conformal Prediction (CCP) (Vovk, 2015; Vovk et al., 2018) strikes a balance between SCP, which can lack efficiency, and full CP, which is computationally intensive (Vovk et al., 2005). CCP can be viewed as a combination of SCP and cross-validation. The data $\mathcal{D}$ is partitioned into $K$ equal folds $\mathcal{D}_1, \ldots, \mathcal{D}_K$ of size $m = |\mathcal{I}|/K$. We assume $\alpha(m+1) \in (1,\infty) \setminus \mathbb{N}$. For each observation with index $i$, the CCP score is computed using a model trained without the fold containing $i$: $s_i = s_{\hat{\mu}(\mathcal{D} \setminus \mathcal{D}_{k(i)})}(X_i, Y_i)$, $i = 1, \ldots, n$, where $\mathcal{D}_{k(i)}$

denotes the fold containing observation $i$.

As formulated in Gasparin & Ramdas (2025), the CCP set is defined as

$$\mathcal{C}^{ccp}(X_{n+1}) := \Big\{ y : \frac{1}{K} \sum_{k=1}^{K} P_k(y) > \alpha + (1-\alpha)\frac{K-1}{K+n} \Big\},$$
(11)

where the fold-wise conformal p-values are given by

$$P_k(y) := \frac{1 + \sum_{i \in \mathcal{I}_k} \mathbf{1}\Big\{ s_{\hat{\mu}(\mathcal{D} \setminus \mathcal{D}_{k(i)})}(X_{n+1}, y) \le s_i \Big\}}{m + 1}.$$
(12)

The CCP set in (11) satisfies the finite-sample coverage guarantee

$$\mathbb{P}\big(Y_{n+1} \in \mathcal{C}^{ccp}(X_{n+1})\big) \ge 1 - 2\alpha - 2\frac{(1-\alpha)(1-1/K)}{n/K+1}.$$
(13)

A key limitation of CCP is its weaker marginal coverage guarantee, even though in practice it often attains empirical coverage close to $1 - \alpha$.

Gasparin & Ramdas (2025) further improve the efficiency of CCP by exploiting the exchangeability of fold-wise p-values to leverage stronger aggregation results (Gasparin et al., 2025). All of their proposed variants come with a $1 - 2\alpha$ coverage guarantee.

Our contribution builds on CCP and its extensions by introducing an e-value–based CCP method that remains valid under arbitrary dependence among the fold-wise p-values, satisfies the $1 - \alpha$ finite-sample coverage guarantee, and preserves the computational complexity of standard CCP. The main idea is to transform the fold-wise p-variables $P_k$ (12) into e-variables using our P2E calibrator:

$$E_k^{ccp}(y) = F_{m,\alpha}(P_k(y)), \; k = 1, \dots, K,$$
(14)

Then, we admissibly merge them by averaging and apply Proposition 1.5 to obtain

$$\mathcal{C}_{ccp}^{\mathbf{ev}}(X_{n+1}) = \Big\{ \frac{1}{K} \sum_{k=1}^{K} E_k^{ccp} < U/\alpha \Big\}.$$
(15)

We refer to this method as **ECCP**. In the special case where the $P_k$, and consequently the $E_k$, are exchangeable, we further propose two variants that leverage Proposition 1.6:

$$\mathcal{C}_{ccp}^{\mathbf{ev}-ex}(X_{n+1}) = \Big\{ \sup_{t \le K} \frac{1}{t} \sum_{k=1}^{t} E_k^{ccp} < 1/\alpha \Big\},$$
(16)

$$\mathcal{C}_{ccp,U}^{\mathbf{ev}-ex}(X_{n+1}) = \mathcal{C}_{ccp}^{\mathbf{ev}-ex}(X_{n+1}) \cap \{ E_1 < U/\alpha \},$$
(17)

with $U \sim \text{Unif}(0,1)$ an independent variable. We refer to these methods as **ECCP-Exch** and **UR-ECCP-Exch**.

**Proposition 4.1.** *If $(X_{n+1}, Y_{n+1})$ is exchangeable with $\mathcal{D}$, the e-CCP set (15) satisfies the finite-sample coverage guarantee in (2). If, in addition, the fold-wise e-values $E_1^{\text{ccp}}, \dots, E_K^{\text{ccp}}$ are exchangeable, then the sets (16) and (17) satisfy (2).*

**E-Conformal Aggregation.** Conformal Aggregation (CA) combines nonconformity scores or the resulting prediction sets from multiple base models into a single prediction set, while retaining finite-sample validity. The e-value framework is thus natural in aggregation problems, because we can use admissible e-merging rules regardless of the dependence of the e-variables.

We adopt the SCP framework with $K > 1$ calibration sets $\mathcal{D}_{cal}^{(1)}, \dots, \mathcal{D}_{cal}^{(K)}$ of sizes $m_1, \dots, m_K$, each associated with a pretrained predictor $\hat{\mu}^1, \dots, \hat{\mu}^K$ and corresponding nonconformity scores $s_1, \dots, s_K$. We assume $\alpha(m_k + 1) \in (1, \infty) \setminus \mathbb{N}, \forall k$. For a new $X_{n+1}$, we compute fold-wise p-values

$$P_k(y) = \frac{1}{m_k + 1}\Big(1 + \sum_{z \in \mathcal{D}_{cal}^{(k)}} \mathbf{1}\{s_k(z) \ge s_k(X_{n+1}, y)\}\Big),$$
(18)

and obtain e-values via our P2E calibrator:

$$E_k(y) = F_{m_k,\alpha}\big(P_k(y)\big), \quad k = 1, \dots, K.$$
(19)

Let $U \sim \text{Unif}(0,1)$ be an independent random variable. By admissibly merging the e-values in (19), we proceed analogously to the CCP construction. To achieve stronger performance, we leverage Wang (2025) and employ weighted aggregation of e-values:

$$\mathcal{C}_{\text{Wagg}}^{\mathbf{ev}}(X_{n+1}) = \Big\{ \sum_{k=1}^{K} \omega_k E_k < \frac{1}{\alpha} \Big\},$$
(20)

where the weights are selected to minimize some criterion, and its randomized version $\mathcal{C}_{\text{Wagg}}^{\mathbf{ev}\text{-}\mathbf{U}}(X_{n+1})$, obtained by replacing the threshold $1/\alpha$ with $U/\alpha$.

To do so, we split each calibration set into two disjoint parts, $\mathcal{S}_{\text{cal}}^{(k)} = \mathcal{S}_{\text{tune}}^{(k)} \sqcup \mathcal{S}_{\text{inf}}^{(k)}$. The tuning split $\{\mathcal{D}_{\text{tune}}^{(k)}\}_{k=1}^{K}$ is used only to select the aggregation weights $\omega^\star = (\omega_1^\star, \dots, \omega_K^\star) \in \Delta_K$, whereas the inference split $\{\mathcal{D}_{\text{inf}}^{(k)}\}_{k=1}^{K}$ is used only to construct the final conformal e-values.

**Proposition 4.2.** *Assuming that, for each $k$, $\mathcal{D}_{\text{inf}}^{(k)}$ and the test point $(X_{n+1}, Y_{n+1})$ are exchangeable, the sets $\mathcal{C}_{Wagg}^{\mathbf{ev}}(X_{n+1}), \mathcal{C}_{Wagg}^{\mathbf{ev}\text{-}\mathbf{U}}(X_{n+1})$ satisfy (2).*

We refer to these methods as **WECA**, and **UR-WECA**, respectively. The core advantage of the WECA method lies in its ability to select *data-dependent* weights that are

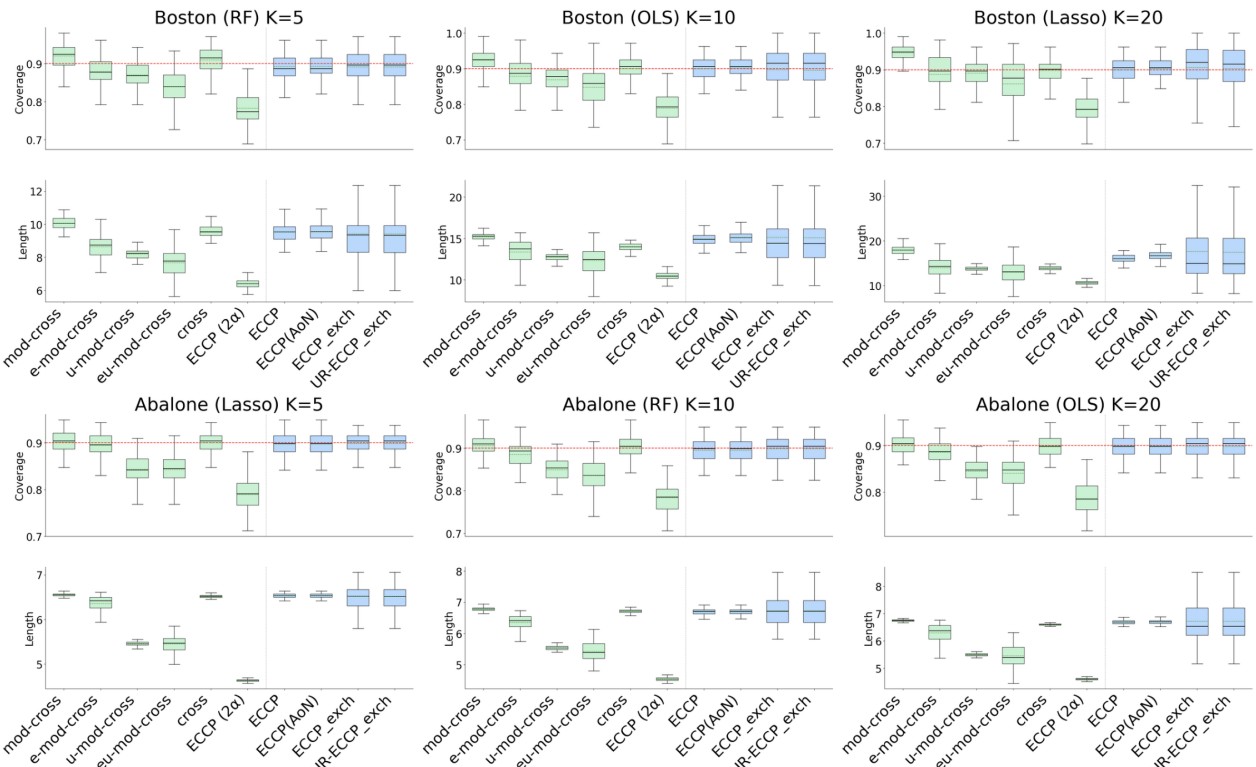

*Figure 2.* Distribution of mean prediction set length and empirical coverage obtained using different regression algorithms and different Cross-CP methods over 100 seeds, for $\alpha = 0.1$

optimized to minimize some type of criterion. In our case, we select the weights to minimize the expected prediction set size. We describe the practical implementation of this method and how coverage holds in Appendix F.1.

## 5. Experiments and Results

We benchmark our methods on OpenML regression datasets (Vanschoren et al., 2014) using an extensive experimental suite that spans a diverse range of base models and multiple CCP configurations[4].

### 5.1. Cross-Conformal Prediction

We follow the experimental design of Gasparin & Ramdas (2025), where CCP p-values are constructed to be exchangeable, using three regressors: Ordinary Least Squares (OLS), Lasso Regression, and Random Forest (RF).

**Methods.** CCP methods provide different coverage guarantees, so for a fair comparison we group methods having the same guarantee:

First, we consider the CCP baseline of Vovk (2015) and

---

[4]https://github.com/Nabil-Ala/P2E_calibration

| Method | Len. | Cov. |
|---|---|---|
| **Abalone** ($K$=10, RF) | | |
| ECCP | $6.70 \pm 0.09$ | $0.90 \pm 0.02$ |
| ECCP($F_1$) | $29.07 \pm 1.93$ | $0.89 \pm 0.03$ |
| ECCP($F_2$) | $10.21 \pm 0.35$ | $0.90 \pm 0.02$ |
| ECCP($F_3$) | $32.02 \pm 1.99$ | $0.89 \pm 0.02$ |
| **Boston** ($K$=10, RF) | | |
| ECCP | $9.76 \pm 0.63$ | $0.89 \pm 0.03$ |
| ECCP($F_1$) | $67.15 \pm 3.99$ | $0.90 \pm 0.03$ |
| ECCP($F_2$) | $53.74 \pm 4.21$ | $0.92 \pm 0.02$ |
| ECCP($F_3$) | $59.18 \pm 4.17$ | $0.90 \pm 0.03$ |

*Table 1.* CCP results with other calibrators. Values are reported as mean $\pm$ standard deviation across 100 random seeds.

the methods with $1 - 2\alpha$ coverage: the variants of Gasparin & Ramdas (2025) (`mod-cross`, `e-mod-cross`, `u-mod-cross`, `eu-mod-cross`), and our ECCP run at level $2\alpha$, denoted ECCP($2\alpha$).

Second, we report the $1 - \alpha$ methods, including our ECCP, ECCP-Exch, and UR-ECCP-Exch. To highlight the role of our P2E calibrator, we also run ECCP with alternative p-to-e calibrators: $F_1(p) = -\log p$, $F_2(p) = p^{-1/2} - 1$, $F_3(p) := 2(1 - p)$ and $F_{\text{AoN}}$.

We denote these variants by ECCP($F_1$), ECCP($F_2$),

*Table 2.* Empirical coverage and prediction set size for conformal aggregation methods. Values are reported as mean $\pm$ standard deviation across 20 random seeds, for $\alpha = 0.05$.

| Method | 361234 | | 361235 | | 361237 | | 361244 | |
|---|---|---|---|---|---|---|---|---|
| | Cov. | Len. | Cov. | Len. | Cov. | Len. | Cov. | Size |
| CM | $0.98 \pm 0.01$ | $3.85 \pm 0.20$ | $0.98 \pm 0.01$ | $3.27 \pm 0.15$ | $0.99 \pm 0.01$ | $2.88 \pm 0.12$ | $0.98 \pm 0.01$ | $6.90 \pm 1.34$ |
| CR | $0.97 \pm 0.01$ | $3.56 \pm 0.19$ | $0.97 \pm 0.01$ | $2.44 \pm 0.17$ | $0.96 \pm 0.02$ | $1.98 \pm 0.15$ | $0.98 \pm 0.01$ | $6.58 \pm 1.28$ |
| $P_{\text{Agg}}$ | $0.98 \pm 0.01$ | $3.91 \pm 0.20$ | $0.99 \pm 0.01$ | $3.33 \pm 0.12$ | $0.99 \pm 0.01$ | $2.97 \pm 0.10$ | $0.98 \pm 0.01$ | $6.91 \pm 1.32$ |
| COLA-S | $0.95 \pm 0.01$ | $2.90 \pm 0.19$ | $0.96 \pm 0.02$ | $1.71 \pm 0.32$ | $0.95 \pm 0.03$ | $1.64 \pm 0.25$ | $0.97 \pm 0.02$ | $5.23 \pm 1.48$ |
| WECA | $0.95 \pm 0.01$ | $2.88 \pm 0.19$ | $0.95 \pm 0.01$ | $1.71 \pm 0.29$ | $0.95 \pm 0.03$ | $1.65 \pm 0.25$ | $0.96 \pm 0.02$ | $5.09 \pm 2.06$ |
| UR-WECA | $0.95 \pm 0.01$ | $2.88 \pm 0.18$ | $0.95 \pm 0.01$ | $1.70 \pm 0.30$ | $0.95 \pm 0.03$ | $1.64 \pm 0.25$ | $0.96 \pm 0.02$ | $4.93 \pm 1.86$ |

| | Method | Cov. | Len. |
|---|---|---|---|
| **361234** | WECA($F_1$) | $1.00 \pm 0.00$ | $72.14 \pm 15.33$ |
| | UR-WECA($F_1$) | $0.95 \pm 0.01$ | $50.36 \pm 10.76$ |
| | WECA($F_2$) | $1.00 \pm 0.00$ | $7.75 \pm 1.40$ |
| | UR-WECA($F_2$) | $0.95 \pm 0.01$ | $4.88 \pm 0.43$ |
| | WECA($F_3$) | $1.00 \pm 0.00$ | $72.14 \pm 15.33$ |
| | UR-WECA($F_3$) | $0.95 \pm 0.01$ | $65.34 \pm 13.92$ |
| **361235** | WECA($F_1$) | $1.00 \pm 0.00$ | $38.43 \pm 2.99$ |
| | UR-WECA($F_1$) | $0.95 \pm 0.01$ | $28.73 \pm 2.48$ |
| | WECA($F_2$) | $1.00 \pm 0.00$ | $38.43 \pm 2.99$ |
| | UR-WECA($F_2$) | $0.96 \pm 0.02$ | $13.98 \pm 1.68$ |
| | WECA($F_3$) | $1.00 \pm 0.00$ | $38.43 \pm 2.99$ |
| | UR-WECA($F_3$) | $0.95 \pm 0.01$ | $34.76 \pm 2.93$ |
| **361237** | WECA($F_1$) | $1.00 \pm 0.00$ | $32.05 \pm 3.24$ |
| | UR-WECA($F_1$) | $0.95 \pm 0.02$ | $24.65 \pm 2.71$ |
| | WECA($F_2$) | $1.00 \pm 0.00$ | $32.05 \pm 3.24$ |
| | UR-WECA($F_2$) | $0.96 \pm 0.01$ | $15.64 \pm 2.07$ |
| | WECA($F_3$) | $1.00 \pm 0.00$ | $32.05 \pm 3.24$ |
| | UR-WECA($F_3$) | $0.95 \pm 0.02$ | $28.89 \pm 2.91$ |
| **361244** | WECA($F_1$) | $1.00 \pm 0.00$ | $115.49 \pm 34.62$ |
| | UR-WECA($F_1$) | $0.97 \pm 0.02$ | $87.96 \pm 25.82$ |
| | WECA($F_2$) | $1.00 \pm 0.00$ | $115.49 \pm 34.62$ |
| | UR-WECA($F_2$) | $0.97 \pm 0.02$ | $54.66 \pm 17.04$ |
| | WECA($F_3$) | $1.00 \pm 0.00$ | $115.49 \pm 34.62$ |
| | UR-WECA($F_3$) | $0.97 \pm 0.02$ | $103.89 \pm 30.73$ |

*Table 3.* Performance of WECA and UR-WECA using alternative p-to-e calibrators: $F_1(p) := -\log(p)$, $F_2(p) := p^{-1/2} - 1$, $F_3(p) := 2(1 - p)$.

$\text{ECCP}(F_3)$, and $\text{ECCP}(\text{AoN})$.

**Results.** We report the results as boxplots in Figure 2. Among the $1 - 2\alpha$ methods (shown in green), $\text{ECCP}(2\alpha)$ is consistently the most efficient: it yields substantially smaller prediction sets compared to CCP and the variants of Gasparin & Ramdas (2025), while achieving empirical coverage close to the target coverage of 80%.

Turning to the $1-\alpha$ methods (in blue), $\text{ECCP}(\text{AoN})$ shows great efficiency, as expected, since $F_{\text{AoN}}$ is set-preserving. This supports the intuition behind our framework. Although we know theoretically that it induces sets larger than ECCP (equation (10)), it asymptotically approaches ECCP's per-

formance as the calibration set size increases (equivalently, as K decreases) as expected. The variants ECCP-Exch and UR-ECCP-Exch also perform well on average, but exhibit higher variability across seeds, likely due to the asymmetric nature of their aggregation rule as pointed out by Gasparin & Ramdas (2025).

Comparing to classic p-to-e calibrators of the literature, $\text{ECCP}(F_1)$, $\text{ECCP}(F_2)$, and $\text{ECCP}(F_3)$, perform substantially worse[5]; we therefore report them separately in Table 1. These results align again with the idea that set-inflating calibrators can lead to noticeably weaker efficiency. Additional experiments are provided in Appendix D.

Finally, we also demonstrate in Appendix C how ECCP can improve one aspect of stability of standard CCP.

### 5.2. Conformal Aggregation

We consider the SCP framework for aggregation in section 4, where we consider $K = 7$ base learners including linear models, tree-based methods, neural networks, and Bayesian regressors. A key advantage of our framework is that WECA can accommodate settings in which different models are calibrated on calibration sets of different sizes. This is more realistic in applications where models may be trained or calibrated using data from different sources. To reflect this setting, we consider heterogeneous calibration sizes by assigning to each predictor a random subset of a larger calibration set; see Appendix F for details.

**Methods.** We compare our proposed methods, **WECA** and **UR-WECA** based on our P2E calibrator, against state-of-the-art aggregation approaches: majority vote (**CM**) and its randomized variant (**CR**) (Gasparin & Ramdas, 2024); a p-value–based approach ($P_{\text{Agg}}$), which averages the conformal p-values per model and applies the threshold at $\alpha/2$ instead of $\alpha$. All the above methods have the $1 - \alpha$ coverage guarantee. Additionally, we implement the recent

---

[5]Average length is computed on a finite bounded evaluation grid. For the alternative calibrators $F_1, F_2, F_3$, the true conformal set may be unbounded (if the aggregated e-variable is always less than $1/\alpha$).

aggregation method **COLA-s** from (Xu et al., 2025), which constructs prediction sets by intersecting conformal sets obtained at optimized miscoverage levels.

**Results.** The results are reported in Table 2. We observe that all methods achieve the finite-sample coverage guarantee, consistent with our theory. CM, CR and $P_{\text{Agg}}$ often over-cover, reflecting their conservative behavior. Our methods are the most efficient among the aggregation methods, highlighting the benefit of the randomization step with $U \sim \text{Unif}(0,1)$ in the UR-WECA method, which consistently produces tighter prediction sets. The role of set-preservation is further supported by Table 3, where we evaluate WECA and UR-WECA under the same setup but using alternative calibrators. Again, using the calibrators $F_1, F_2, F_3$ produce overly conservative prediction sets.

Overall, these results indicate that the gains of our approach come from the combination of the proposed P2E calibration with admissible e-value merging, further enhanced by randomization.

More broadly, our e-CP framework is modular and can be extended to additional settings and applications; we discuss possible future directions in Appendix G.

## 6. Conclusion

We present an e-value perspective on CP by calibrating the conformal p-variable into an e-variable. We show that most classical p-to-e calibrators inflate the SCP set, motivating set-preservation as a key structural property for efficiency. Building on this insight, we introduce a principled new P2E calibrator that is set-preserving. This construction unlocks the benefits of e-value theory and randomization results that we apply to enhance CA and CCP. Our experiments support the set-preserving intuition and demonstrate consistent improvements in efficiency while maintaining coverage, providing a practical step toward efficient and robust e-value based distribution-free uncertainty quantification.

## Impact Statement

This work advances the field of reliable uncertainty quantification and machine learning. By providing mathematically grounded frameworks that enhance trust and transparency, our approach facilitates the deployment of practical, accessible models tailored for real-world applications. Although our framework is domain-agnostic, it should not be applied in settings that enable harmful, unethical, or discriminatory uses.

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

# A. Comparison of e-CP from (Balinsky & Balinsky, 2024) and p-CP sets

Many p-CP methods yield sharp prediction sets, whereas e-CP methods offer greater flexibility and stronger theoretical guarantees through e-value theory. Our objective is to preserve the practical efficiency of p-CP while harnessing the theoretical advantages of e-CP. To evaluate this trade-off, we compare the resulting prediction set sizes, providing insight into the efficiency gap between the two approaches.

Recall that SCP operates on a set of calibration scores $s_1, \cdots, s_n$ and a new test point $X_{n+1}$, under the standard CP exchangeability assumption. Assuming the scores are positive, Balinsky & Balinsky (2024) proposed an e-variable defined as

$$E_n(y) = \frac{(n+1)\, s_{\hat{\mu}}(X_{n+1}, y)}{\sum_{i=1}^n s_i + s_{\hat{\mu}}(X_{n+1}, y)}, \qquad y \in \mathcal{Y}. \tag{21}$$

So the corresponding e-CP set is given by

$$\mathcal{C}^{\mathrm{ev}}(X_{n+1}) = \big\{ y \in \mathcal{Y} : E_n(y) < 1/\alpha \big\}, \tag{22}$$

and it enjoys finite-sample marginal coverage by Proposition 1.2. We will refer to this set as the e-CP(Balinsky).

This design offers two practical advantages:

1. Unlike p-CP, which compares $s_{\hat{\mu}}(X_{n+1}, y)$ to the empirical $(1-\alpha)$ quantile $s_{(\lceil(1-\alpha)(n+1)\rceil)}$, e-CP requires no sorting at prediction time. We simply evaluate the e-variable for each candidate label.

2. It provides a principled choice of e-variable for CP, which can be readily applied or adapted to various settings (see, e.g., Gauthier et al. (2025)).

Let us compare the prediction set (22) and the classic p-CP set. Denoting $S = \sum_{i=1}^n s_i$, we can rewrite (22) as

$$\mathcal{C}^{\mathrm{ev}}(X_{n+1}) = \Big\{ y \in \mathcal{Y} : \; s_{\hat{\mu}}(X_{n+1}, y) \; < \; \frac{S}{\alpha(n+1) - 1} \Big\}. \tag{23}$$

Recall that the corresponding p-CP set is given by

$$\mathcal{C}^{\mathbf{pv}}(X_{n+1}) = \big\{ y \in \mathcal{Y} : s_{\hat{\mu}}(X_{n+1}, y) \leq s_{(k)} \big\}, \tag{24}$$

where $k = \lceil (1-\alpha)(n+1) \rceil$.

We note that

$$k = \lceil (1-\alpha)(n+1) \rceil \; \Rightarrow \; k - 1 < (1-\alpha)(n+1),$$

which implies

$$n - k + 1 \; > \; n - (1-\alpha)(n+1) \; = \; \alpha(n+1) - 1.$$

Since at least the top $n - k + 1$ scores are each $\geq s_{(k)}$, it follows that

$$S \; \geq \; \sum_{i=k}^n s_{(i)} \; \geq \; (n - k + 1)\, s_{(k)} \; > \; \big(\alpha(n+1) - 1\big)\, s_{(k)}.$$

Dividing by $\alpha(n+1) - 1$ (which is assumed to be $> 0$) yields

$$\frac{S}{\alpha(n+1) - 1} \; > \; s_{(k)} \quad \Longrightarrow \quad \mathcal{C}^{\mathbf{pv}}(X_{n+1}) \subseteq \mathcal{C}^{\mathbf{ev}}(X_{n+1}).$$

Therefore, the p-CP set is always contained within the e-CP set for any target coverage level. In practice, this difference can be substantial. This observation underscores the main limitation of the e-variable defined in (21), which tends to produce larger prediction sets. In contrast, our proposed e-variable in (9) retains the sharpness of the p-CP construction while operating within the e-value framework.

To illustrate this point, we conduct a small experiment comparing the prediction sets in (23) and (24) on both regression and classification datasets from OpenML. We consider three base models: Linear Regression, Random Forest (RF), and a Multi-Layer Perceptron (MLP). Each dataset is split into 60% for training, 20% for calibration, and 20% for testing. We report the mean prediction set length and empirical coverage in Tables 4 and 5. We observe that p-CP achieves coverage close to the nominal level, whereas e-CP(Balinsky) consistently over-covers and produces substantially larger prediction sets.

| Dataset | Model | P-CP | | E-CP(Balinsky) | |
|---|---|---|---|---|---|
| | | Cov | Len | Cov | Len |
| boston | Linear | $0.884 \pm 0.055$ | $14.495 \pm 3.083$ | $1.000 \pm 0.000$ | $74.592 \pm 7.153$ |
| boston | MLP | $0.909 \pm 0.037$ | $12.678 \pm 1.568$ | $1.000 \pm 0.000$ | $66.596 \pm 7.998$ |
| boston | RF | $0.914 \pm 0.020$ | $10.613 \pm 0.805$ | $0.997 \pm 0.006$ | $50.816 \pm 3.965$ |
| california | Linear | $0.901 \pm 0.005$ | $2.194 \pm 0.043$ | $1.000 \pm 0.000$ | $10.717 \pm 0.103$ |
| california | MLP | $0.901 \pm 0.007$ | $1.613 \pm 0.034$ | $1.000 \pm 0.000$ | $7.282 \pm 0.166$ |
| california | RF | $0.899 \pm 0.007$ | $1.562 \pm 0.051$ | $1.000 \pm 0.000$ | $6.746 \pm 0.153$ |
| kin8nm | Linear | $0.904 \pm 0.011$ | $0.651 \pm 0.012$ | $1.000 \pm 0.000$ | $3.283 \pm 0.063$ |
| kin8nm | MLP | $0.908 \pm 0.004$ | $0.269 \pm 0.006$ | $1.000 \pm 0.000$ | $1.284 \pm 0.025$ |
| kin8nm | RF | $0.904 \pm 0.016$ | $0.478 \pm 0.014$ | $1.000 \pm 0.000$ | $2.330 \pm 0.059$ |

*Table 4.* Empirical mean prediction set size and coverage for 3 regression datasets (across 10 seeds, $\alpha = 0.1$).

| Dataset | Model | P-CP | | E-CP(Balinsky) | |
|---|---|---|---|---|---|
| | | Cov | Len | Cov | Len |
| breast_cancer | Linear | $0.924 \pm 0.030$ | $0.93 \pm 0.03$ | $0.986 \pm 0.010$ | $1.01 \pm 0.02$ |
| breast_cancer | MLP | $0.911 \pm 0.037$ | $0.94 \pm 0.05$ | $1.000 \pm 0.000$ | $1.91 \pm 0.26$ |
| breast_cancer | RF | $0.916 \pm 0.035$ | $0.94 \pm 0.04$ | $0.999 \pm 0.003$ | $1.51 \pm 0.40$ |
| cifar10 | Linear | $0.896 \pm 0.013$ | $7.27 \pm 0.14$ | $1.000 \pm 0.000$ | $10.00 \pm 0.00$ |
| cifar10 | MLP | $0.902 \pm 0.010$ | $5.75 \pm 0.19$ | $1.000 \pm 0.000$ | $10.00 \pm 0.00$ |
| cifar10 | RF | $0.893 \pm 0.012$ | $5.09 \pm 0.14$ | $1.000 \pm 0.000$ | $10.00 \pm 0.00$ |
| mnist_784 | Linear | $0.906 \pm 0.005$ | $1.07 \pm 0.03$ | $1.000 \pm 0.000$ | $10.00 \pm 0.00$ |
| mnist_784 | MLP | $0.903 \pm 0.008$ | $0.97 \pm 0.01$ | $1.000 \pm 0.000$ | $10.00 \pm 0.00$ |
| mnist_784 | RF | $0.900 \pm 0.008$ | $0.95 \pm 0.01$ | $1.000 \pm 0.000$ | $10.00 \pm 0.00$ |

*Table 5.* Empirical mean prediction set size and coverage for 3 classification datasets (across 10 seeds, $\alpha = 0.1$).

The following proposition characterizes the asymptotic ratio between the lengths of the e-CP(Balinsky) and p-CP prediction sets.

**Proposition A.1.** *Assume the absolute residual calibration scores are i.i.d. and nonnegative, with $\mu := \mathbb{E}[s_1] \in (0, \infty)$ and $\sigma^2 := \mathrm{Var}(s_1) < \infty$. Let $F$ denote their cdf, and assume $F$ is continuous at $q := F^{-1}(1 - \alpha)$ with density $f(q) > 0$. Then, as $n \to \infty$,*

$$\frac{|\mathcal{C}^{\mathbf{ev}}(X_{n+1})|}{|\mathcal{C}^{\mathbf{pv}}(X_{n+1})|} = \frac{S/((n+1)\alpha - 1)}{s_{(k)}} \longrightarrow \frac{\mu}{\alpha q} \geq 1.$$

*Proof.* Let $S = \sum_{i=1}^{n} s_i$ and $s_{(k)}$ be the $k$th order statistic with $k = \lceil (n+1)(1-\alpha) \rceil$. By the strong law of large numbers, $S/n \to \mu$, and $((n+1)\alpha - 1)/n \to \alpha$, hence $S/((n+1)\alpha - 1) \to \mu/\alpha$ a.s. By Glivenko–Cantelli and continuity of $F$ at $q := F^{-1}(1 - \alpha)$ with $f(q) > 0$, we have $s_{(k)} \to q$ a.s. The continuous mapping theorem yields $\frac{S/((n+1)\alpha - 1)}{s_{(k)}} \xrightarrow{\text{a.s.}} \frac{\mu}{\alpha q}$. This ratio is greater than one because $\mathbb{E}[s_1] \geq t \, \mathbb{P}(s_1 \geq t)$ by MI, and by continuity at $q$, $\mathbb{P}(s_1 \geq q) = \alpha$, so $\mu = \mathbb{E}[s_1] \geq q \, \alpha$. $\square$

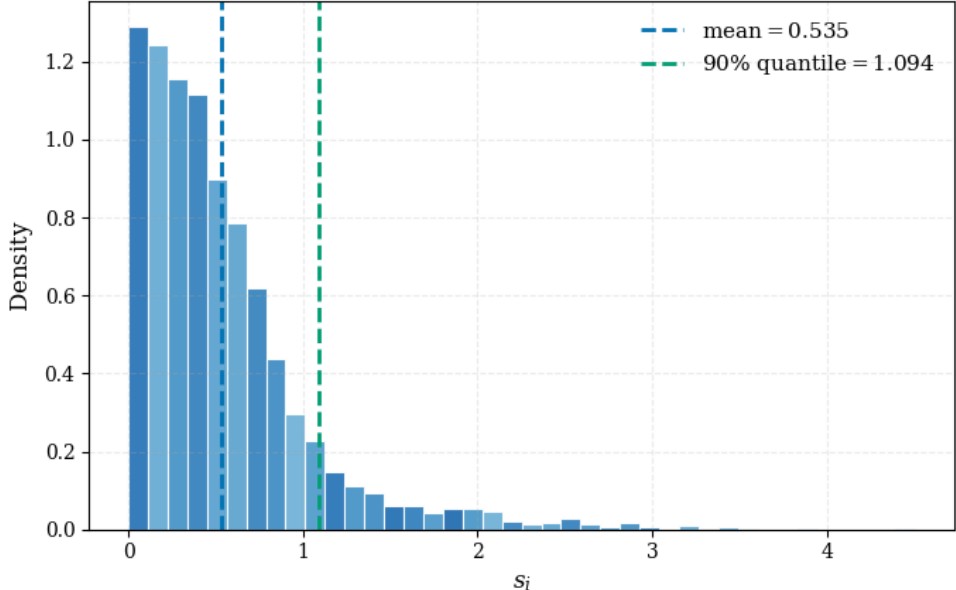

*Figure 3.* Empirical distribution of calibration scores (California Housing) for Linear Regression model.

Our empirical findings are consistent with the theoretical result above. For instance, for the Linear Regression model, the *California housing* dataset (Table 4), the ratio of the average interval lengths between the two sets is $10.717/2.194 \approx 4.884$. Figure 3 shows the empirical distribution of the calibration scores for Linear Regression model, together with the estimated mean and $(1 - \alpha)$-quantile. We observe that

$$\frac{\hat{\mu}}{\alpha \hat{q}} = \frac{0.535}{0.1 \times 1.094} \approx 4.885,$$

which matches the theoretical prediction and explains the substantial over-coverage.

# B. Proofs of results

We assume in all the following that $\alpha(n+1) \notin \mathbb{N}$ and $\alpha(n+1) > 1$.

## B.1. Set-inflation of p-to-e calibrators

P-values and e-values can be converted into one another by means of *p-to-e calibrators* (Vovk & Wang, 2021; Ramdas & Wang, 2025) :

**Definition B.1.** A p-to-e calibrator is a decreasing function $f : [0, \infty) \to [0, \infty]$ such that, for any p-variable $P$ for any hypothesis $\mathcal{H}$, the random variable $f(P)$ is an e-variable for $\mathcal{H}$.

A useful characterization of the class of p-to-e calibrators is given in the following proposition.

**Proposition B.2.** *A decreasing function* $f : [0, \infty) \to [0, \infty]$ *with* $f = 0$ *on* $(1, \infty)$ *is a p-to-e calibrator if and only if* $\int_0^1 f(p)\, dp \leq 1$.

However, we show that (almost) no p-to-e calibrator is set-preserving:

### Proof of Proposition 2.3

*Proof.* This comes from the fact that $f(p) \leq 1/p$ for all $p$ when $f$ is a p-to-e calibrator. Indeed, if $\exists p^* \in (0, 1)$ such that $f(p^*) > 1/p^*$, then $\int_0^1 f(q)\, dq \geq p^* f(p^*) > 1$, thus violating the condition in Proposition B.2. Therefore, for any p-to-e calibrator $f$,

$$\mathcal{C}^{\mathbf{pv}}(X_{n+1}) \subseteq \{ f(P_n) < 1/\alpha \}.$$

Thus,

$$|\mathcal{C}^{\mathbf{pv}}(X_{n+1})| \leq |\mathcal{C}^{\mathbf{ev}}(X_{n+1})|$$

for any p-to-e calibrator $f$. Moreover, it is straightforward to see that $F_{\text{AoN}}$ is set-preserving, by definition of this calibrator.

Now to prove that only the AoN calibrator is set-preserving, assume that $F$ is a p-to-e calibrator, left-continuous at $\alpha$, and set-preserving for every nonconformity score used to define $P_n$, i.e.

$$\{y : P_n(y) > \alpha\} = \{y : F(P_n(y)) < 1/\alpha\}$$

for all $n$, and all score functions. Since $F$ is a p-to-e calibrator, $F(p) \leq 1/p$, hence $F(\alpha) \leq 1/\alpha$. Suppose $F(\alpha) < 1/\alpha$. By left-continuity, choose $\varepsilon > 0$ such that $F(p) < 1/\alpha$ for $p \in (\alpha - \varepsilon, \alpha]$. Pick $N = n + 1$ large with $q_N = \lfloor \alpha N \rfloor / N \in (\alpha - \varepsilon, \alpha]$. Because set-preservation is assumed for every score, consider the absolute-residual score $s(x, y) = |y - \mu(x)|$, assuming distinct calibration scores; choosing $y_0$ so that exactly $\lfloor \alpha N \rfloor - 1$ calibration scores at least as large as $s(X_{n+1}, y_0)$ gives $P_n(y_0) = q_N$. Then $q_N \leq \alpha$, so $y_0 \notin \{P_n > \alpha\}$, but $F(P_n(y_0)) < 1/\alpha$, so $y_0 \in \{F(P_n) < 1/\alpha\}$, contradiction. Thus $F(\alpha) = 1/\alpha$. Since $F$ is decreasing, $F(p) \geq 1/\alpha$ for $p \leq \alpha$, and $\int_0^1 F \leq 1$ forces $F(p) = 1/\alpha$ on $(0, \alpha]$ and $F(p) = 0$ on $(\alpha, 1]$. Therefore $F = F_{\text{AoN}}$.

$\square$

## B.2. Proof of Proposition 2.4

**Proposition.** *Let* $n \geq 1$, *and let* $P_n$ *be the p-variable in* (3). *Then, for any* $\alpha \in (0, 1)$, *there exists a function* $f_{n,\alpha} \in \mathcal{E}$ *such that*

$$\mathbb{E}_{\mathbb{H}_0}\left( \frac{f_{n,\alpha}(P_n)}{\alpha f_{n,\alpha}(\alpha)} \right) = 1, \tag{25}$$

*under the null hypothesis* $\mathbb{H}_0 : Y_{n+1} = y$.

*Proof.* Fix $\alpha \in (0, 1)$ and $n \geq 1$. Under the null the p-variable is uniformly distributed on $\{\frac{1}{n+1}, \cdots, \frac{n}{n+1}, 1\}$. Thus, for $f \in \mathcal{E}$, condition (25) is equivalent to

$$\frac{1}{n+1} \sum_{k=1}^{n+1} f\left( \frac{k}{n+1} \right) = \alpha f(\alpha). \tag{$\star$}$$

We will need the following lemma to complete the proof.

**Lemma B.3.** *Let $V \in \mathbb{R}^d$ have at least one positive and at least one negative coordinate. Then there exists $x \in \mathbb{R}^d_{>0}$ such that $V \cdot x = 0$.*

*Proof.* Take any $x^{(0)} \in \mathbb{R}^d_{>0}$ and set $s := V \cdot x^{(0)}$. If $s = 0$, the claim holds. If $s > 0$, choose an index $i$ such that $V_i < 0$ and define $t := s/(-V_i) > 0$. Then $x := x^{(0)} + te_i \in \mathbb{R}^d_{>0}$, and $V \cdot x = s + tV_i = s - s = 0$. If $s < 0$, the same argument applies with an index $j$ satisfying $V_j > 0$. $\square$

Let $N := n + 1$ and $j := \lfloor \alpha N \rfloor$, so that

$$\frac{j}{N} < \alpha < \frac{j+1}{N}.$$

Define the augmented grid

$$y_1 = \frac{1}{N}, \ldots, y_j = \frac{j}{N}, \quad y_{j+1} = \alpha, \quad y_{j+2} = \frac{j+1}{N}, \ldots, y_{N+1} = 1,$$

and set $u_k = f(y_k)$. Let

$$\delta_m := u_m - u_{m+1}, \qquad m = 1, \ldots, N, \qquad c := u_{N+1}.$$

Then

$$u_k = c + \sum_{m=k}^{N} \delta_m, \qquad k = 1, \ldots, N.$$

Since the point $y_{j+1} = \alpha$ is not part of the conformal grid, condition $(\star)$ is equivalent to

$$\frac{1}{N} \left( \sum_{k=1}^{j} u_k + \sum_{k=j+2}^{N+1} u_k \right) = \alpha u_{j+1}.$$

Substituting the representation of the $u_k$'s gives after simplification

$$N(1-\alpha)c + \sum_{m=1}^{j} m\delta_m + (j - \alpha N)\delta_{j+1} + \sum_{m=j+2}^{N} (m - 1 - \alpha N)\delta_m = 0.$$

Define

$$\beta_0 := N(1 - \alpha), \qquad \beta_m := m, \quad m = 1, \ldots, j,$$

$$\beta_{j+1} := j - \alpha N < 0, \qquad \beta_m := m - 1 - \alpha N, \quad m = j + 2, \ldots, N.$$

Then $\beta_0 > 0$, $\beta_{j+1} < 0$, and $\beta_m > 0$ for all $m \neq j + 1$. By Lemma B.3, there exists

$$(c, \delta_1, \ldots, \delta_N) \in \mathbb{R}^{N+1}_{>0}$$

such that

$$\beta_0 c + \sum_{m=1}^{N} \beta_m \delta_m = 0.$$

Thus $u_1 > \cdots > u_{N+1} > 0$. Finally, choose any $u_0 > u_1$ at $y_0 = 0$, and define $f$ by linear interpolation through

$$(y_0, u_0), (y_1, u_1), \ldots, (y_{N+1}, u_{N+1}).$$

Then $f$ is positive and strictly decreasing on $[0, 1]$, and by construction it satisfies $(\star)$.

$\square$

*Remark* B.4. When $\alpha = \frac{j}{n+1}$ with $1 \leq j \leq n$, no solution exists. Indeed, writing $u_k := f\left(\frac{k}{n+1}\right)$, and $u_1 > \cdots > u_{n+1} > 0$, equation $(\star)$ becomes

$$\sum_{k=1}^{n+1} u_k = j u_j \quad \Longleftrightarrow \quad \sum_{k \neq j} u_k = (j-1) u_j. \tag{26}$$

But strict monotonicity implies $\sum_{k=1}^{j-1} u_k > (j-1) u_j$ and $\sum_{k=j+1}^{n+1} u_k > 0$, thus leading to a contradiction.

**Proof of Proposition 2.5** Let $f_n \in \mathcal{E}$ be positive and strictly decreasing, satisfying

$$\frac{1}{n+1} \sum_{k=1}^{n+1} f_n\left(\frac{k}{n+1}\right) = \alpha f_n(\alpha). \tag{$\star$}$$

Then

$$\frac{f_n(p)}{f_n(\alpha)} \longrightarrow \mathbf{1}_{\{p \le \alpha\}}$$

for every fixed $p \in (0, 1]$.

*Proof.* By definition we have

$$0 \le \alpha(n+1) - \lfloor \alpha(n+1) \rfloor < 1.$$

Multiplying $(\star)$ by $n + 1$, we get

$$\sum_{k=1}^{n+1} f_n\left(\frac{k}{n+1}\right) = \alpha(n+1) f_n(\alpha).$$

Subtracting $\lfloor \alpha(n+1) \rfloor f_n(\alpha)$ from both sides gives

$$\sum_{k=1}^{\lfloor \alpha(n+1) \rfloor} \left[ f_n\left(\frac{k}{n+1}\right) - f_n(\alpha) \right] + \sum_{k=\lfloor \alpha(n+1) \rfloor + 1}^{n+1} f_n\left(\frac{k}{n+1}\right) = \left(\alpha(n+1) - \lfloor \alpha(n+1) \rfloor\right) f_n(\alpha). \tag{1}$$

Both terms on the left-hand side are nonnegative. Indeed, if $k \le m_n$, then $k/(n+1) \le \alpha$, so by monotonicity, $f_n\left(\frac{k}{n+1}\right) \ge f_n(\alpha)$.

- Now fix $p < \alpha$. We prove that

$$\frac{f_n(p)}{f_n(\alpha)} \to 1.$$

Since $f_n$ is decreasing and $p < \alpha$, we already have $\frac{f_n(p)}{f_n(\alpha)} \ge 1$. For every integer $k \le \lfloor p(n+1) \rfloor$, we have

$$\frac{k}{n+1} \le p < \alpha \implies f_n\left(\frac{k}{n+1}\right) \ge f_n(p).$$

Therefore,

$$f_n\left(\frac{k}{n+1}\right) - f_n(\alpha) \ge f_n(p) - f_n(\alpha).$$

Summing over $k = 1, \dots, \lfloor p(n+1) \rfloor$, we get

$$\sum_{k=1}^{\lfloor p(n+1) \rfloor} \left[ f_n\left(\frac{k}{n+1}\right) - f_n(\alpha) \right] \ge \lfloor p(n+1) \rfloor \left[ f_n(p) - f_n(\alpha) \right].$$

The left-hand side is bounded above by the first sum in (1), and the first sum in (1) is bounded above by the right-hand side of (1). Hence

$$\lfloor p(n+1) \rfloor \left[ f_n(p) - f_n(\alpha) \right] \le \left(\alpha(n+1) - \lfloor \alpha(n+1) \rfloor\right) f_n(\alpha).$$

Since

$$0 \le \alpha(n+1) - \lfloor \alpha(n+1) \rfloor < 1,$$

we obtain

$$0 \le \frac{f_n(p)}{f_n(\alpha)} - 1 \le \frac{1}{\lfloor p(n+1) \rfloor}.$$

Now considering an increasing sequence $n_k$, with $\alpha(n_k + 1) \notin \mathbb{N}$, with $n_k \to \infty, k \to \infty$, the right-hand side goes to 0. Therefore,

$$\frac{f_n(p)}{f_n(\alpha)} \to 1.$$

- Fix $p > \alpha$. We now prove that $\frac{f_n(p)}{f_n(\alpha)} \to 0$. For a sufficiently large $n$, consider $k$ s.t.

$$\alpha < \frac{k}{n+1} \leq p \implies f_n\left(\frac{k}{n+1}\right) \geq f_n(p).$$

Therefore,

$$\sum_{k=\lfloor \alpha(n+1)\rfloor+1}^{\lfloor p(n+1)\rfloor} f_n\left(\frac{k}{n+1}\right) \geq (\lfloor p(n+1)\rfloor - \lfloor \alpha(n+1)\rfloor)\, f_n(p).$$

The left-hand side is bounded above by the second sum in (1), and the second sum in (1) is bounded above by the right-hand side of (1). Hence

$$(\lfloor p(n+1)\rfloor - \lfloor \alpha(n+1)\rfloor)\, f_n(p) \leq (\alpha(n+1) - \lfloor \alpha(n+1)\rfloor)\, f_n(\alpha).$$

Since

$$0 \leq \alpha(n+1) - \lfloor \alpha(n+1)\rfloor < 1,$$

we obtain

$$0 \leq \frac{f_n(p)}{f_n(\alpha)} \leq \frac{1}{\lfloor p(n+1)\rfloor - \lfloor \alpha(n+1)\rfloor}.$$

Because $p > \alpha$, we have $\lfloor p(n+1)\rfloor - \lfloor \alpha(n+1)\rfloor \to \infty$. Hence

$$\frac{f_n(p)}{f_n(\alpha)} \to 0.$$

- Finally, if $p = \alpha$, then trivially $\frac{f_n(\alpha)}{f_n(\alpha)} = 1$.

Combining the three cases, we conclude that, for every fixed $p \in (0,1]$,

$$\frac{f_n(p)}{f_n(\alpha)} \longrightarrow \mathbf{1}_{\{p \leq \alpha\}}.$$

### B.3. Proof of Theorem 2.6

**Theorem.** *Let $n \geq 1$ and $\alpha \in (0,1)$, and let $P_n$ be the p-variable defined in (3). Then, for any $s \in \left(\alpha, \frac{\lceil \alpha(n+1)\rceil}{n+1}\right)$, there exists a constant $C_{n,\alpha} > 0$ such that*

$$E_{n,\alpha} = \frac{1}{\alpha} \cdot \frac{1 + \exp(C_{n,\alpha}(\alpha - s))}{1 + \exp(C_{n,\alpha}(P_n - s))} \tag{27}$$

*is an exact e-variable.*

*Proof.* Define

$$\mathcal{L}(C,p) := \frac{1}{\alpha} \cdot \frac{1 + \exp(C(\alpha - s))}{1 + \exp(C(p - s))}, \qquad C > 0,\ p > 0.$$

For each fixed $C > 0$, the mapping $p \mapsto \mathcal{L}(C,p)$ is strictly decreasing and positive. Hence, we seek $C > 0$ such that

$$\mathbb{E}_{\mathbb{H}_0}(E_{n,\alpha}) = 1 \iff \Sigma(C) := \frac{1}{n+1}\sum_{k=1}^{n+1} \mathcal{L}\left(C, \frac{k}{n+1}\right) = 1. \tag{28}$$

The function $\Sigma(C)$ is continuous on $(0, \infty)$. We examine its limiting behavior:

- As $C \to 0^+$, we have $\exp(C(\alpha - s)) \to 1$, and thus each term of the sum tends to $1/\alpha$; consequently,

$$\lim_{C \to 0^+} \Sigma(C) = \tfrac{1}{\alpha} > 1.$$

- As $C \to \infty$, $\exp(C(\alpha - s)) \to 0$, and

$$\frac{1}{1 + \exp(C(\frac{k}{n+1} - s))} \longrightarrow \begin{cases} 1, & \frac{k}{n+1} < s, \\ 0, & \frac{k}{n+1} > s. \end{cases}$$

Hence,

$$\lim_{C \to \infty} \Sigma(C) = \frac{1}{\alpha} \cdot \frac{\#\{k : \frac{k}{n+1} < s\}}{n+1} = \frac{\lfloor \alpha(n+1) \rfloor}{\alpha(n+1)} < 1.$$

By continuity of $\Sigma$, the Intermediate Value Theorem guarantees the existence of a $C_{n,\alpha} > 0$ such that $\Sigma(C_{n,\alpha}) = 1$.

**Properties of our P2E calibrator.** For fixed $n, \alpha$, our P2E calibrator

$$F_{n,\alpha}(p) := \frac{1}{\alpha} \cdot \frac{1 + \exp(C_{n,\alpha}(\alpha - s))}{1 + \exp(C_{n,\alpha}(p - s))}$$

is smooth thanks to its sigmoid-like form. Noting that $C_{n,\alpha} > 0$, it is strictly decreasing as a function of $p$, since

$$\frac{\partial}{\partial p} F_{n,\alpha}(p) = -\frac{C_{n,\alpha}}{\alpha} \frac{(1 + \exp(C_{n,\alpha}(\alpha - s))) \exp(C_{n,\alpha}(p - s))}{(1 + \exp(C_{n,\alpha}(p - s)))^2} < 0.$$

Moreover, it satisfies $F_{n,\alpha}(\alpha) = \frac{1}{\alpha}$. Its inverse is obtained explicitly as follows. For $e = F_{n,\alpha}(p)$,

$$p = F_{n,\alpha}^{-1}(e) = s + \frac{1}{C_{n,\alpha}} \log\left(\frac{1 + \exp(C_{n,\alpha}(\alpha - s))}{\alpha e} - 1\right),$$

for $e$ in the range of $F_{n,\alpha}$.

**Dominance Over AoN.** For

$$F_{n,\alpha}(p) = \frac{1}{\alpha} \frac{1 + \exp(C_{n,\alpha}(\alpha - s))}{1 + \exp(C_{n,\alpha}(p - s))}$$

with $s > \alpha$:

- If $p \le \alpha$, then $p - s \le \alpha - s$, so

$$\exp(C(p - s)) \le \exp(C(\alpha - s)),$$

hence the denominator is no larger than the numerator and

$$F_{n,\alpha}(p) \ge 1/\alpha = F_{\text{AoN}}(p).$$

- If $p > \alpha$, then $F_{\text{AoN}}(p) = 0$, while $F_{n,\alpha}(p) > 0$.

Therefore $F_{n,\alpha} \ge F_{\text{AoN}}$ pointwise.

$\square$

### B.4. Proof sketch of Proposition 4.1

It follows directly from the Markov-type bounds of Subsection 1.3. Applying MI (1.2) or UR-MI (1.5) then yields the stated finite-sample coverage guarantees.

For the exchangeable constructions of ECCP-Exch and UR-ECCP-Exch, the corresponding results follow by invoking the exchangeable Markov inequality (EMI) in (1.6).

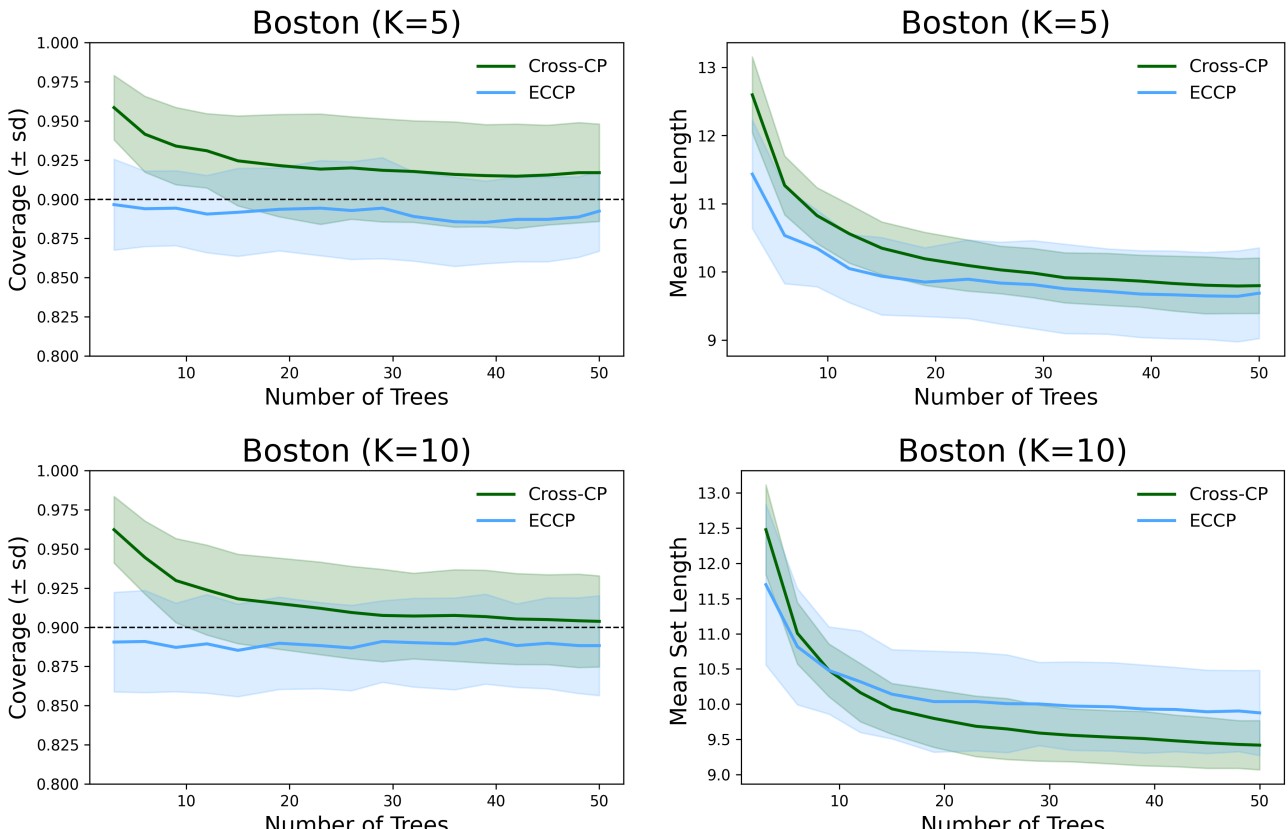

*Figure 4.* Evolution of coverage and mean prediction set size on the Boston dataset as a function of the number of trees in the RF model, for both CCP (green) and ECCP (blue), averaged over 25 seeds. The surrounding shaded area indicates the range of mean $\pm$ standard deviation. Miscoverage level is set to $\alpha = 0.1$.

## C. ECCP improves the stability of CCP

Empirical studies confirm that CCP can either under-cover or remain overly conservative depending on the stability of the underlying model (Linusson et al., 2017). To illustrate this, Figure 4 shows how coverage and prediction set size evolve with the number of trees in a Random Forest (RF) when using CCP and ECCP. With few trees, when the RF is unstable, CCP becomes markedly conservative, in line with Linusson et al. (2017). As the number of trees increases and the RF stabilizes, CCP's coverage becomes normal again. In contrast, ECCP consistently remains near the target coverage regardless of the stability of the RF model.

## D. ECCP Additional experiments

### D.1. ECCP methods

We provide additional experimental results for the CCP methods on Boston , Abalone, and Parkinson (Tsanas & Little, 2009) datasets, for three base models (OLS, RF, and Lasso) as in (Gasparin & Ramdas, 2025).

*Table 6.* Comparison of prediction set size and coverage, Boston dataset, $K = 15$, across 100 random seeds.

| Base | Metric | CCP | e-mod-cross | u-mod-cross | eu-mod-cross | ECCP($2\alpha$) |
|---|---|---|---|---|---|---|
| OLS | Size | $13.970 \pm 0.456$ | $14.100 \pm 1.720$ | $13.288 \pm 0.439$ | $12.569 \pm 1.955$ | $10.665 \pm 0.473$ |
| | Cov. | $0.897 \pm 0.032$ | $0.895 \pm 0.055$ | $0.880 \pm 0.034$ | $0.854 \pm 0.067$ | $0.793 \pm 0.042$ |
| RF | Size | $9.140 \pm 0.357$ | $9.208 \pm 1.450$ | $8.615 \pm 0.323$ | $7.994 \pm 1.618$ | $6.549 \pm 0.317$ |
| | Cov. | $0.902 \pm 0.035$ | $0.894 \pm 0.049$ | $0.882 \pm 0.034$ | $0.849 \pm 0.070$ | $0.784 \pm 0.046$ |
| Lasso | Size | $13.926 \pm 0.463$ | $14.109 \pm 1.736$ | $13.268 \pm 0.447$ | $12.548 \pm 2.006$ | $10.632 \pm 0.480$ |
| | Cov. | $0.898 \pm 0.032$ | $0.895 \pm 0.053$ | $0.881 \pm 0.032$ | $0.854 \pm 0.066$ | $0.792 \pm 0.043$ |

| Base | Metric | ECCP | ECCP($F_{\text{AoN}}$) | ECCP($F_1$) | ECCP($F_2$) | ECCP($F_3$) |
|---|---|---|---|---|---|---|
| OLS | Size | $15.458 \pm 0.850$ | $18.727 \pm 1.300$ | $65.724 \pm 3.768$ | $71.783 \pm 3.746$ | $63.206 \pm 3.979$ |
| | Cov. | $0.905 \pm 0.033$ | $0.926 \pm 0.029$ | $0.926 \pm 0.029$ | $0.907 \pm 0.031$ | $0.908 \pm 0.031$ |
| RF | Size | $10.126 \pm 0.616$ | $12.662 \pm 1.153$ | $63.487 \pm 4.036$ | $70.362 \pm 3.932$ | $61.264 \pm 4.204$ |
| | Cov. | $0.899 \pm 0.032$ | $0.922 \pm 0.027$ | $0.921 \pm 0.024$ | $0.907 \pm 0.028$ | $0.903 \pm 0.026$ |
| Lasso | Size | $15.479 \pm 0.839$ | $18.786 \pm 1.309$ | $65.746 \pm 3.768$ | $71.786 \pm 3.745$ | $63.204 \pm 3.984$ |
| | Cov. | $0.903 \pm 0.031$ | $0.926 \pm 0.028$ | $0.926 \pm 0.029$ | $0.907 \pm 0.032$ | $0.908 \pm 0.030$ |

*Table 7.* Comparison of prediction set size and coverage, Abalone dataset, $K = 15$, across 100 random seeds.

| Base | Metric | CCP | e-mod-cross | u-mod-cross | eu-mod-cross | ECCP($2\alpha$) |
|---|---|---|---|---|---|---|
| OLS | Size | $6.603 \pm 0.034$ | $6.365 \pm 0.300$ | $5.477 \pm 0.050$ | $5.444 \pm 0.324$ | $4.600 \pm 0.036$ |
| | Cov. | $0.898 \pm 0.022$ | $0.888 \pm 0.027$ | $0.844 \pm 0.027$ | $0.840 \pm 0.034$ | $0.787 \pm 0.033$ |
| RF | Size | $6.696 \pm 0.057$ | $6.358 \pm 0.282$ | $5.547 \pm 0.064$ | $5.441 \pm 0.399$ | $4.557 \pm 0.063$ |
| | Cov. | $0.901 \pm 0.024$ | $0.884 \pm 0.029$ | $0.848 \pm 0.029$ | $0.838 \pm 0.038$ | $0.783 \pm 0.036$ |
| Lasso | Size | $6.513 \pm 0.035$ | $6.302 \pm 0.255$ | $5.497 \pm 0.046$ | $5.472 \pm 0.309$ | $4.659 \pm 0.036$ |
| | Cov. | $0.900 \pm 0.022$ | $0.891 \pm 0.025$ | $0.846 \pm 0.029$ | $0.844 \pm 0.034$ | $0.790 \pm 0.032$ |

| Base | Metric | ECCP | ECCP($F_{\text{AoN}}$) | ECCP($F_1$) | ECCP($F_2$) | ECCP($F_3$) |
|---|---|---|---|---|---|---|
| OLS | Size | $6.716 \pm 0.076$ | $6.755 \pm 0.080$ | $10.350 \pm 0.327$ | $30.647 \pm 1.956$ | $32.181 \pm 1.975$ |
| | Cov. | $0.898 \pm 0.021$ | $0.899 \pm 0.021$ | $0.901 \pm 0.022$ | $0.891 \pm 0.024$ | $0.891 \pm 0.023$ |
| RF | Size | $6.747 \pm 0.087$ | $6.788 \pm 0.086$ | $10.294 \pm 0.329$ | $30.578 \pm 1.958$ | $32.133 \pm 1.976$ |
| | Cov. | $0.897 \pm 0.024$ | $0.899 \pm 0.024$ | $0.904 \pm 0.025$ | $0.893 \pm 0.026$ | $0.893 \pm 0.024$ |
| Lasso | Size | $6.665 \pm 0.082$ | $6.704 \pm 0.083$ | $10.670 \pm 0.340$ | $30.772 \pm 1.946$ | $32.233 \pm 1.971$ |
| | Cov. | $0.899 \pm 0.022$ | $0.900 \pm 0.022$ | $0.902 \pm 0.024$ | $0.892 \pm 0.024$ | $0.891 \pm 0.024$ |

*Table 8.* Comparison of prediction set size and coverage, Parkinsons dataset, $K = 20$, across 100 random seeds.

| Base | Metric | CCP | e-mod-cross | u-mod-cross | eu-mod-cross | ECCP($2\alpha$) |
|------|--------|-----|-------------|-------------|--------------|-----------------|
| OLS | Size | $29.342 \pm 0.429$ | $28.462 \pm 1.244$ | $26.239 \pm 0.256$ | $26.087 \pm 1.418$ | $23.462 \pm 0.236$ |
|     | Cov. | $0.897 \pm 0.008$ | $0.886 \pm 0.019$ | $0.850 \pm 0.009$ | $0.846 \pm 0.027$ | $0.792 \pm 0.010$ |
| RF | Size | $5.537 \pm 0.426$ | $5.014 \pm 0.600$ | $4.226 \pm 0.348$ | $3.996 \pm 0.663$ | $2.978 \pm 0.282$ |
|    | Cov. | $0.898 \pm 0.008$ | $0.877 \pm 0.018$ | $0.851 \pm 0.008$ | $0.830 \pm 0.030$ | $0.772 \pm 0.010$ |
| Lasso | Size | $29.330 \pm 0.417$ | $28.492 \pm 1.213$ | $26.243 \pm 0.254$ | $26.088 \pm 1.415$ | $23.468 \pm 0.237$ |
|       | Cov. | $0.897 \pm 0.008$ | $0.886 \pm 0.018$ | $0.850 \pm 0.009$ | $0.846 \pm 0.027$ | $0.793 \pm 0.010$ |

| Base | Metric | ECCP | ECCP($F_{\text{AoN}}$) | ECCP($F_1$) | ECCP($F_2$) | ECCP($F_3$) |
|------|--------|------|------------------------|-------------|-------------|-------------|
| OLS | Size | $30.100 \pm 0.528$ | $30.154 \pm 0.532$ | $39.431 \pm 0.335$ | $70.983 \pm 0.667$ | $70.004 \pm 0.688$ |
|     | Cov. | $0.897 \pm 0.008$ | $0.898 \pm 0.008$ | $0.910 \pm 0.005$ | $0.900 \pm 0.005$ | $0.899 \pm 0.005$ |
| RF | Size | $5.627 \pm 0.422$ | $5.648 \pm 0.423$ | $13.145 \pm 0.748$ | $59.823 \pm 0.853$ | $59.351 \pm 0.880$ |
|    | Cov. | $0.890 \pm 0.009$ | $0.891 \pm 0.009$ | $0.898 \pm 0.007$ | $0.885 \pm 0.007$ | $0.884 \pm 0.006$ |
| Lasso | Size | $30.093 \pm 0.524$ | $30.147 \pm 0.527$ | $39.438 \pm 0.338$ | $70.983 \pm 0.667$ | $70.006 \pm 0.688$ |
|       | Cov. | $0.898 \pm 0.008$ | $0.898 \pm 0.008$ | $0.910 \pm 0.005$ | $0.900 \pm 0.005$ | $0.899 \pm 0.005$ |

Overall, among the $1 - 2\alpha$ methods, our ECCP is the most efficient, consistently outperforming the competing baselines. For the $1 - \alpha$ methods, we observe the same pattern as before: ECCP comes with a large-sample performance guarantee of $F_{AoN}$ and remains substantially more efficient than alternatives based on set-inflating calibrators, such as $F_1(p) = -\log p$, $F_2(p) = p^{-1/2} - 1$, and $F_3(p) = 2(1 - p)$.

## E. Details on the P2E calibrator

### E.1. The $\alpha(n + 1) \notin \mathbb{N}$ condition

Our framework is based on the assumption $\alpha(n + 1) \notin \mathbb{N}$, which makes our main theorem valid. This assumption is only a mild technical condition. It is used to ensure that the interval $\left(\alpha, \frac{\lceil \alpha(n+1) \rceil}{n+1}\right)$ is nonempty, so that the parameter $s$ in the P2E calibrator can be chosen strictly larger than $\alpha$. For example, for $\alpha = 0.1$, it occurs for $n = 9, 19, 29, \ldots$. Additionally, if $\alpha$ is irrational, then this case never happens, although in practice we usually consider $\alpha$ to be $0.1, 0.05, 0.025$, etc. Since this case occurs for a condition only dependent on the calibration size and the miscoverage level, it can be avoided simply, by removing or adding one calibration point. Indeed, if $\alpha(n + 1) = k \in \mathbb{N}$, then $\alpha n = k - \alpha \notin \mathbb{N}$ and $\alpha(n + 2) = k + \alpha \notin \mathbb{N}$ for any $\alpha \in (0, 1)$.

### E.2. Sensitivity to the choice of $s$.

We study the effect of the tuning parameter $s \in \left(\alpha, \frac{\lceil \alpha(n+1) \rceil}{n+1}\right)$ on the performance of our methods:

$$s_w = w\alpha + (1 - w)\frac{\lceil \alpha(n + 1) \rceil}{n + 1} \qquad \text{for } w \in [0, 1].$$

and report in the tables below the average prediction set length and empirical coverage vary for different values of $w$ for the three base predictors we consider.

We observe that the performance varies only slightly across the different values of $w$, and hence across the corresponding choices of $s_w = w\alpha + (1 - w)\frac{\lceil \alpha(n+1) \rceil}{n+1} \in \left(\alpha, \frac{\lceil \alpha(n+1) \rceil}{n+1}\right)$. This suggests that the tuning parameter $s$ has little practical impact on the transformed e-variable produced by our P2E calibrator, and consequently on both the efficiency and the coverage. This stability can also be understood from the sigmoid form of the calibrator in (9): geometrically, small changes in the location parameter $s$ induce only mild changes in the shape of the map $p \to F_{n,\alpha}(p)$.

*Table 9.* Mean length and coverage for OLS across 30 seeds for different weights, Boston dataset, $\alpha = 0.1$, $K = 5$.

| $[w, 1-w]$ | ECCP | | ECCP-exch | | UR-ECCP-exch | |
|---|---|---|---|---|---|---|
| | Len. | Cov. | Len. | Cov. | Len. | Cov. |
| $[0.05, 0.95]$ | $14.351 \pm 0.564$ | $0.895 \pm 0.026$ | $14.898 \pm 2.172$ | $0.900 \pm 0.050$ | $14.811 \pm 2.123$ | $0.898 \pm 0.051$ |
| $[0.1, 0.9]$ | $14.351 \pm 0.564$ | $0.895 \pm 0.026$ | $14.898 \pm 2.172$ | $0.900 \pm 0.050$ | $14.811 \pm 2.123$ | $0.898 \pm 0.051$ |
| $[0.2, 0.8]$ | $14.351 \pm 0.565$ | $0.895 \pm 0.026$ | $14.898 \pm 2.172$ | $0.900 \pm 0.050$ | $14.812 \pm 2.124$ | $0.898 \pm 0.051$ |
| $[0.3, 0.7]$ | $14.359 \pm 0.570$ | $0.895 \pm 0.026$ | $14.898 \pm 2.172$ | $0.900 \pm 0.050$ | $14.831 \pm 2.134$ | $0.898 \pm 0.051$ |
| $[0.4, 0.6]$ | $14.376 \pm 0.571$ | $0.894 \pm 0.025$ | $14.898 \pm 2.172$ | $0.900 \pm 0.050$ | $14.863 \pm 2.153$ | $0.900 \pm 0.050$ |
| $[0.5, 0.5]$ | $14.384 \pm 0.573$ | $0.895 \pm 0.025$ | $14.898 \pm 2.172$ | $0.900 \pm 0.050$ | $14.888 \pm 2.170$ | $0.900 \pm 0.050$ |
| $[0.6, 0.4]$ | $14.389 \pm 0.574$ | $0.895 \pm 0.025$ | $14.898 \pm 2.172$ | $0.900 \pm 0.050$ | $14.897 \pm 2.172$ | $0.900 \pm 0.050$ |
| $[0.7, 0.3]$ | $14.389 \pm 0.574$ | $0.895 \pm 0.025$ | $14.898 \pm 2.172$ | $0.900 \pm 0.050$ | $14.898 \pm 2.172$ | $0.900 \pm 0.050$ |
| $[0.8, 0.2]$ | $14.388 \pm 0.575$ | $0.895 \pm 0.025$ | $14.898 \pm 2.172$ | $0.900 \pm 0.050$ | $14.898 \pm 2.172$ | $0.900 \pm 0.050$ |
| $[0.9, 0.1]$ | $14.388 \pm 0.574$ | $0.895 \pm 0.025$ | $14.898 \pm 2.172$ | $0.900 \pm 0.050$ | $14.898 \pm 2.172$ | $0.900 \pm 0.050$ |

*Table 10.* Mean length and coverage for RF across 30 seeds for different weights, Boston dataset, $\alpha = 0.1$, $K = 5$.

| $[w, 1-w]$ | ECCP | | ECCP-exch | | UR-ECCP-exch | |
|---|---|---|---|---|---|---|
| | Len. | Cov. | Len. | Cov. | Len. | Cov. |
| $[0.05, 0.95]$ | $9.439 \pm 0.659$ | $0.884 \pm 0.031$ | $9.509 \pm 1.530$ | $0.886 \pm 0.044$ | $9.456 \pm 1.502$ | $0.884 \pm 0.043$ |
| $[0.1, 0.9]$ | $9.439 \pm 0.659$ | $0.884 \pm 0.031$ | $9.509 \pm 1.530$ | $0.886 \pm 0.044$ | $9.456 \pm 1.502$ | $0.884 \pm 0.043$ |
| $[0.2, 0.8]$ | $9.439 \pm 0.659$ | $0.884 \pm 0.031$ | $9.509 \pm 1.530$ | $0.886 \pm 0.044$ | $9.457 \pm 1.502$ | $0.884 \pm 0.043$ |
| $[0.3, 0.7]$ | $9.444 \pm 0.663$ | $0.884 \pm 0.031$ | $9.509 \pm 1.530$ | $0.886 \pm 0.044$ | $9.472 \pm 1.513$ | $0.885 \pm 0.043$ |
| $[0.4, 0.6]$ | $9.457 \pm 0.669$ | $0.884 \pm 0.031$ | $9.509 \pm 1.530$ | $0.886 \pm 0.044$ | $9.488 \pm 1.520$ | $0.885 \pm 0.043$ |
| $[0.5, 0.5]$ | $9.468 \pm 0.674$ | $0.885 \pm 0.032$ | $9.509 \pm 1.530$ | $0.886 \pm 0.044$ | $9.502 \pm 1.529$ | $0.886 \pm 0.044$ |
| $[0.6, 0.4]$ | $9.469 \pm 0.676$ | $0.884 \pm 0.031$ | $9.509 \pm 1.530$ | $0.886 \pm 0.044$ | $9.508 \pm 1.530$ | $0.886 \pm 0.044$ |
| $[0.7, 0.3]$ | $9.469 \pm 0.677$ | $0.884 \pm 0.032$ | $9.509 \pm 1.530$ | $0.886 \pm 0.044$ | $9.509 \pm 1.530$ | $0.886 \pm 0.044$ |
| $[0.8, 0.2]$ | $9.469 \pm 0.677$ | $0.884 \pm 0.032$ | $9.509 \pm 1.530$ | $0.886 \pm 0.044$ | $9.509 \pm 1.530$ | $0.886 \pm 0.044$ |
| $[0.9, 0.1]$ | $9.469 \pm 0.676$ | $0.884 \pm 0.032$ | $9.509 \pm 1.530$ | $0.886 \pm 0.044$ | $9.509 \pm 1.530$ | $0.886 \pm 0.044$ |

*Table 11.* Mean length and coverage for Lasso across 30 seeds for different weights, Boston dataset, $\alpha = 0.1$, $K = 5$.

| $[w, 1-w]$ | ECCP | | ECCP-exch | | UR-ECCP-exch | |
|---|---|---|---|---|---|---|
| | Len. | Cov. | Len. | Cov. | Len. | Cov. |
| $[0.05, 0.95]$ | $14.393 \pm 0.576$ | $0.896 \pm 0.026$ | $14.887 \pm 2.108$ | $0.903 \pm 0.047$ | $14.803 \pm 2.078$ | $0.901 \pm 0.051$ |
| $[0.1, 0.9]$ | $14.393 \pm 0.576$ | $0.896 \pm 0.026$ | $14.887 \pm 2.108$ | $0.903 \pm 0.047$ | $14.803 \pm 2.078$ | $0.901 \pm 0.051$ |
| $[0.2, 0.8]$ | $14.394 \pm 0.577$ | $0.896 \pm 0.026$ | $14.887 \pm 2.108$ | $0.903 \pm 0.047$ | $14.803 \pm 2.079$ | $0.901 \pm 0.051$ |
| $[0.3, 0.7]$ | $14.405 \pm 0.580$ | $0.896 \pm 0.025$ | $14.887 \pm 2.108$ | $0.903 \pm 0.047$ | $14.820 \pm 2.084$ | $0.901 \pm 0.051$ |
| $[0.4, 0.6]$ | $14.423 \pm 0.582$ | $0.896 \pm 0.025$ | $14.887 \pm 2.108$ | $0.903 \pm 0.047$ | $14.854 \pm 2.098$ | $0.902 \pm 0.048$ |
| $[0.5, 0.5]$ | $14.433 \pm 0.582$ | $0.896 \pm 0.026$ | $14.887 \pm 2.108$ | $0.903 \pm 0.047$ | $14.877 \pm 2.110$ | $0.903 \pm 0.049$ |
| $[0.6, 0.4]$ | $14.437 \pm 0.582$ | $0.896 \pm 0.026$ | $14.887 \pm 2.108$ | $0.903 \pm 0.047$ | $14.886 \pm 2.109$ | $0.903 \pm 0.047$ |
| $[0.7, 0.3]$ | $14.437 \pm 0.582$ | $0.896 \pm 0.026$ | $14.887 \pm 2.108$ | $0.903 \pm 0.047$ | $14.887 \pm 2.109$ | $0.903 \pm 0.047$ |
| $[0.8, 0.2]$ | $14.436 \pm 0.583$ | $0.896 \pm 0.026$ | $14.887 \pm 2.108$ | $0.903 \pm 0.047$ | $14.887 \pm 2.108$ | $0.903 \pm 0.047$ |
| $[0.9, 0.1]$ | $14.436 \pm 0.583$ | $0.896 \pm 0.026$ | $14.887 \pm 2.108$ | $0.903 \pm 0.047$ | $14.887 \pm 2.108$ | $0.903 \pm 0.047$ |

# F. CA Experiments & Details

**Experimental setup.** For each dataset and random seed, we split the data into $50\%$ training, $35\%$ calibration, and $15\%$ test sets. A collection of $K = 7$ regression models $\{\widehat{\mu}_k\}_{k=1}^{K}$ is trained on the same training set: Linear Regression, regularized Lasso, iterative SGDRegressor, a probabilistic model (Bayesian Ridge) and tree-based ensembles (Random

Forest and HistGradientBoosting), and a neural network (MLPRegressor). To study heterogeneous calibration sizes, each model $k$ is assigned its own calibration subset $S_{\text{cal}}^{(k)} \subseteq S_{\text{cal}}$, sampled uniformly without replacement from the common calibration pool, with a random size between $60\%$ and $100\%$ of $|S_{\text{cal}}|$. For model $k$, conformity scores are computed as $R_j^{(k)} = |Y_j - \widehat{\mu}_k(X_j)|, (X_j, Y_j) \in S_{\text{cal}}^{(k)}$. Given a test point $x$ and candidate response $y$, the model-specific conformal p-value is

$$p_k(y) = \frac{1 + \sum_{j \in S_{\text{cal}}^{(k)}} \mathbf{1}\{R_j^{(k)} \geq |y - \widehat{\mu}_k(x)|\}}{|S_{\text{cal}}^{(k)}| + 1},$$

which is calibrated into an e-variable using the calibrators we consider in section 5. All methods are evaluated at nominal miscoverage level $\alpha = 0.05$ over multiple random seeds. Coverage is computed on the test responses, while average prediction-set length is approximated on a finite grid over a data-adaptive interval around the model predictions.

### F.1. Theoretical Validity of WECA

Although, by design, all of our e-value-based methods satisfy the coverage guarantee, we explain here how we implement our WECA method in order for the coverage guaranty to be preserved. The core idea is to choose weights that are independent of a calibration data split, so that the aggregated e-variable

$$E^\omega := \sum_{k=1}^{K} \omega_k E_k$$

yields prediction sets with small average length.

Consider $K$ predictors indexed by $k \in [K]$. Each predictor $k$ may have its own calibration index set, denoted by $I_{\text{cal}}^{(k)}$. For a set of calibration indices $J_k \subseteq I_{\text{cal}}^{(k)}$, let $P_k^{J_k}(x, y)$ denote the conformal p-variable constructed from predictor $k$ using only the calibration data indexed by $J_k$, evaluated at test input $x$ and candidate label $y \in \mathcal{Y}$. Its calibrated e-variable is

$$E_k^{J_k}(x, y) := F_{|J_k|, \alpha}\big(P_k^{J_k}(x, y)\big),$$

where $F_{|J_k|, \alpha}$ is the P2E calibrator corresponding to calibration size $|J_k|$ and miscoverage level $\alpha$.

Denote the space of weight vectors $\omega = (\omega_1, \ldots, \omega_K)$ by the simplex

$$\Delta_K := \left\{ \omega \in \mathbb{R}_+^K : \sum_{k=1}^{K} \omega_k = 1 \right\}.$$

For any $\omega \in \Delta_K$ and any vector of calibration index sets $\mathbf{J} = (J_1, \ldots, J_K)$, define the weighted e-variable

$$E_\omega^{\mathbf{J}}(x, y) := \sum_{k=1}^{K} \omega_k E_k^{J_k}(x, y),$$

and the corresponding prediction set

$$\mathcal{C}_\alpha^{\omega, \mathbf{J}}(x) := \left\{ y \in \mathcal{Y} : E_\omega^{\mathbf{J}}(x, y) < \frac{1}{\alpha} \right\}.$$

We split the calibration data into a *tuning* part, used only to select the weights, and an *inference* part, used only to produce the final conformal set:
$$I_{\text{cal}} = I_{\text{tune}} \sqcup I_{\text{inf}}, \qquad |I_{\text{tune}}| = n_{\text{tune}}, \qquad |I_{\text{inf}}| = n_{\text{inf}}.$$

We then split the tuning indices again:

$$I_{\text{tune}} = I_{\text{tune},1} \sqcup I_{\text{tune},2}, \qquad |I_{\text{tune},1}| = n_{\text{tune},1}, \qquad |I_{\text{tune},2}| = n_{\text{tune},2}.$$

Since each model may have its own calibration set, we define the model-specific splits

$$J_{k,\text{tune},1} := I_{\text{cal}}^{(k)} \cap I_{\text{tune},1}, \qquad J_{k,\text{inf}} := I_{\text{cal}}^{(k)} \cap I_{\text{inf}},$$

with sizes

$$n_{k,\text{tune},1} := |J_{k,\text{tune},1}|, \qquad n_{k,\inf} := |J_{k,\inf}|.$$

We also write

$$\mathbf{J}_{\text{tune},1} := (J_{1,\text{tune},1}, \ldots, J_{K,\text{tune},1}), \qquad \mathbf{J}_{\inf} := (J_{1,\inf}, \ldots, J_{K,\inf}).$$

The first tuning split $I_{\text{tune},1}$ is used as a calibration set to construct the e-variables, while the second tuning split $I_{\text{tune},2}$ is used only to evaluate the empirical prediction-set size. We choose

$$\omega^\star \in \arg\min_{\omega \in \Delta_K} \frac{1}{n_{\text{tune},2}} \sum_{i \in I_{\text{tune},2}} \left| \mathcal{C}_\alpha^{\omega,\mathbf{J}_{\text{tune},1}}(X_i) \right|.$$

After the weights have been selected, the final WECA set is constructed using only the inference calibration split:

$$\mathcal{C}_\alpha^{\text{WECA}}(X_{n+1}) := \left\{ y \in \mathcal{Y} : E_{\inf}(X_{n+1}, y) < \frac{1}{\alpha} \right\},$$

where

$$E_{\inf}(X_{n+1}, y) := E_{\omega^\star}^{\mathbf{J}_{\inf}}(X_{n+1}, y) = \sum_{k=1}^{K} \omega_k^\star E_k^{J_{k,\inf}}(X_{n+1}, y),$$

with

$$E_k^{J_{k,\inf}}(X_{n+1}, y) = F_{n_{k,\inf},\alpha}\left( P_k^{J_{k,\inf}}(X_{n+1}, y) \right).$$

We now show that the final statistic is an e-variable, thanks to the independence of $\omega^*$ from the inference split and the test point:

$$\mathbb{E}_{\mathbb{H}_0}[E_{\inf}] = \sum_{k=1}^{K} \mathbb{E}_{\mathbb{H}_0}\left[ \omega_k^* E_k^{J_{k,\inf}}(X_{n+1}, Y_{n+1}) \right] = \sum_{k=1}^{K} \mathbb{E}_{\mathbb{H}_0}[\omega_k^*] \underbrace{\mathbb{E}_{\mathbb{H}_0}\left[ E_k^{J_{k,\inf}}(X_{n+1}, Y_{n+1}) \right]}_{=1} = \sum_{k=1}^{K} \mathbb{E}_{\mathbb{H}_0}[\omega_k^*] = 1.$$

This proves that WECA preserves the finite-sample coverage guarantee. The randomized version UR-WECA is obtained by replacing the threshold $1/\alpha$ with $U/\alpha$, where $U \sim \text{Unif}(0,1)$ is independent of all data. Its validity follows analogously from the UR-MI.

## G. Discussion and Future Directions

Our methodology enables a simple and general conversion of conformal p-variables into e-variables using any P2E calibrator. Once e-values are obtained, our framework naturally lends itself to several extensions and applications. Beyond the settings discussed above, we highlight a few potential directions for future research.

**Post-hoc selection.** E-values naturally support post-hoc (data-adaptive) inference. Following Koning (2023), a nonnegative random variable $P$ is *post-hoc valid* if, for every (possibly data-dependent) random threshold $\delta > 0$,

$$\sup_{\delta > 0} \mathbb{E}\left[ \frac{\mathbf{1}\{P \leq \delta\}}{\delta} \right] \leq 1.$$

Koning further shows that this property holds if and only if $1/P$ is an e-variable. In our notation, for fixed $\alpha \in (0,1)$ and $n \geq 1$, the post-hoc validity condition can equivalently be written as

$$\sup_{\delta > 0} \mathbb{E}\left[ \frac{\mathbf{1}\{E_{n,\alpha} \geq 1/\delta\}}{\delta} \right] \leq 1,$$

allowing data-dependent, post-hoc choices of $\delta$.

**Randomized e-CP.** As previously discussed, both p-CP and e-CP prediction sets coincide under P2E calibration:

$$\{P_n > \alpha\} = \{E_{n,\alpha} < 1/\alpha\},$$

thus preserving the conformal coverage guarantee. One can further consider a randomized variant using an independent $U \sim \mathrm{Unif}(0, 1)$, where a candidate is included if $E_n < U/\alpha$. This rule defines a prediction set that is nested within the nonrandomized version and remains marginally valid by the UR-MI property:

$$\{E_{n,\alpha} < U/\alpha\} \subset \{E_{n,\alpha} < 1/\alpha\} = \{P_n > \alpha\},$$

with all satisfying the $1 - \alpha$ coverage guarantee. Exploring how such randomization affects the distribution of coverage and the efficiency of the resulting prediction sets represents an interesting direction for future work.

