# OpenReview forum: "Set-Preserving Calibration from Conformal P-Values to E-Values"
_ICML.cc/2026/Conference — ICML 2026 regular_

### Official Review · Reviewer_5dr6 · 2026-02-27

**Soundness:** 3
**Presentation:** 4
**Significance:** 4
**Originality:** 4
**Overall Recommendation:** 6
**Confidence:** 5

**Summary:**

This paper studies how to convert conformal p-values into e-values in a way that avoids the efficiency loss that often occurs when applying standard p-to-e calibrators in conformal prediction. The key idea is to construct a novel set-preserving P2E calibrator, which is also smooth and invertible and strictly positive. Then the authors apply this idea to cross-conformal prediction and conformal aggregation with thorough experiment evaluation.

**Compliance With Llm Reviewing Policy:**

Affirmed.

**Final Justification:**

I recommend this paper for strong acceptance. The paper introduces a new construction of conformal e-values that is both methodologically novel and easy to implement in practice. I find this to be a strong contribution because it is not only theoretically interesting, but also practically useful for applications such as model aggregation and cross-validation-based conformal interval construction. The paper is well motivated, clearly written, and technically solid. In weighing the paper’s strengths and weaknesses, I find its originality, practical significance, and clarity to be clear strengths, and I believe the overall contribution is strong. My main concern in the initial review related to the experimental evaluation, but the authors’ rebuttal fully addressed this issue and increased my confidence in the empirical support for the paper. Overall, the rebuttal reinforced my positive assessment, and I therefore strongly recommend acceptance.

**Key Questions For Authors:**

**Q1.** Line 204-210. The sentence “This justification gives intuition on why why set-preserving calibrators can yield tight prediction sets” reads a bit strange given the preceding inequality. Would it be more accurate to phrase this as “yield tight aggregated prediction sets.” ?

**Q2.** In the experiments, I find ECCP appears to undercover in some settings, which seems inconsistent with the claimed finite-sample
$1-\alpha $  coverage guarantee. Could the authors clarify why this occurs? Is the observed undercoverage attributable to Monte Carlo variability due to a limited number of repetitions? If so, it would be helpful to to increase the number of repetitions (e.g., to 100) to verify the guarantee empirically if computation permits.

**Q3.** Could the authors comment on the computational burden of the proposed P2E calibration in practice? In particular, does computing or searching for $F_{n,\alpha} $ add nontrivial overhead relative to standard conformal baselines, and how does runtime scale with
$n $  and the number of test points?

**Limitations:**

A minor limitation is the computational overhead of the proposed procedure, which is not discussed in the paper.

**Strengths And Weaknesses:**

**Strength 1**: The paper is well organized and strongly motivated. The proposed P2E calibrator is derived in a principled way from the notion of set preservation, and the resulting construction is intuitive and technically sound. Overall, this provides a useful and conceptually clean tool for converting conformal p-values into conformal e-values without inflating prediction sets.

**Strength 2**: The e-value reformulation leads to exact finite-sample coverage and improved efficiency for practically important settings, including cross-conformal prediction and conformal aggregation. This substantially increases the practical relevance of the contribution, since these are common workflows where dependence across folds/splits can invalidate naive p-value aggregation.

**Weakness 1**: The empirical benchmarks for conformal aggregation appear somewhat limited. I encourage the authors to include additional baselines to better substantiate the advantages of the proposed approach. In particular, there is a related line of work on aggregation via data splitting or full conformalization (e.g., [1–3]) as well as extensions of majority-vote style aggregation (e.g., [4]) that also achieve finite-sample validity after aggregation. Even a partial empirical comparison (or a more detailed discussion if experiments are not feasible) would strengthen the evidence for superiority and clarify when the proposed method is preferable.


[1] Yang, Y., & Kuchibhotla, A. K. (2025). Selection and aggregation of conformal prediction sets. Journal of the American Statistical Association, 120(549), 435-447.

[2] Luo, R., & Zhou, Z. (2025). Conformity score averaging for classification. In Forty-second International Conference on Machine Learning.

[3] Xu, C., Yu, Y., Ren, H., Wang, Z., & Zou, C. (2025). Aggregating Conformal Prediction Sets via {\alpha}-Allocation. arXiv preprint arXiv:2511.12065.

[4] Qin, S., He, J., Kuang, Q., Gang, B., & Xia, Y. (2024). Data-light Uncertainty Set Merging with Admissibility. arXiv preprint arXiv:2410.12201.

---

> ### Author Rebuttal · Authors · 2026-03-31
>
> We thank you for your positive and constructive feedback on our work. Please find below our response to your questions.
>
> "*The empirical benchmarks for conformal aggregation appear somewhat limited. I encourage the authors to include additional baselines ... Even a partial empirical comparison (or a more detailed discussion if experiments are not feasible) would strengthen the evidence for superiority and clarify when the proposed method is preferable*".
>
> We thank you for mentioning these additional references, some of which we were not aware of, particularly [4]. We discuss each of them below:
>
> - [1] is classic reference in conformal selection, and is better viewed as a selection baseline than a true aggregation method, since it chooses one conformal predictor from a family rather than combining scores or sets. [2] is closer in spirit to our method because it learns weights to combine multiple score functions, although the paper works in  classification scenario and not directly designed for heterogeneous calibration sizes across predictors.
> - Among the recent set-aggregation methods, [3] is the closest comparison: in particular, COLA-s uses sample splitting and enjoys finite-sample coverage. They propose other variants based full-CP or localized CP. involving a kernel, but both are different from the Split-CP setup we consider. [4] proposes a broader framework, SAT, that can be seen as generalization of Majority vote procedure.
>
> Additionally, we could not find public code for [3,4]. However, we will consider adding some of these baselines in our CA experimental setup in the revised manuscript. We thank the reviewer for mentioning these points, and we will add these relevant references and discuss them in the revised manuscript.
>
> "*The sentence “This justification gives intuition on why why set-preserving calibrators can yield tight prediction sets” reads a bit strange given the preceding inequality. Would it be more accurate to phrase this as “yield tight aggregated prediction sets.”?*
>
> - Your comment is indeed accurate. We will replace the sentence with: *“This justification gives intuition for why set-preserving calibrators, in the context of aggregation, can yield tight aggregated prediction sets.”*
>
> "*In the experiments, I find ECCP appears to undercover in some settings, which seems inconsistent with the claimed finite-sample coverage guarantee..*"
>
> In the box plots of Fig. 2, the Ex-ECCP variants enlarge the overall scale of the figure, which can make these differences harder to see. We therefore report below the empirical coverage of the ECCP method for three models, so as not to overload the table:
>
> ### Table 1. ECCP coverage, $\alpha=0.1$ (Boston dataset)
>
> | K | Model | ECCP Coverage |
> |---|---|---|
> | 5  | OLS   | 0.896 ± 0.028 |
> | 5  | Lasso | 0.899 ± 0.028 |
> | 10 | OLS   | 0.893 ± 0.030 |
> | 10 | RF    | 0.887 ± 0.028 |
> | 20 | OLS   | 0.901 ± 0.031 |
> | 20 | Lasso | 0.899 ± 0.030 |
>
> ### Table 2. ECCP coverage, $\alpha=0.1$ (Abalone dataset)
>
> | K | Model | ECCP Coverage |
> |---|---|---|
> | 5  | OLS   | 0.894 ± 0.019 |
> | 5  | RF    | 0.897 ± 0.021 |
> | 15 | RF    | 0.897 ± 0.023 |
> | 15 | Lasso | 0.895 ± 0.020 |
> | 20 | OLS   | 0.895 ± 0.020 |
> | 20 | Lasso | 0.895 ± 0.017 |
>
> These coverage results are consistent across our 20 random seeds and support the stability of the ECCP method. In particular, they do not indicate undercoverage, which is in line with the theory.
>
> "*Could the authors comment on the computational burden of the proposed P2E calibration in practice? In particular, does computing or searching for $F$ add nontrivial overhead relative to standard conformal baselines, and how does runtime scale with $n$ and the number of test points?*"
>
> - We refer you to our response to reviewer d6hc, who raised a similar issue regarding how we implement the P2E calibrator.
>
> We thank you again for your appreciation of our work as well as for your constructive comments. We hope we have addressed your concerns.
>
> ---
>
> [1] Yang, Y., & Kuchibhotla, A. K. (2025). Selection and aggregation of conformal prediction sets. Journal of the American Statistical Association, 120(549), 435-447.
>
> [2] Luo, R., & Zhou, Z. (2025). Conformity score averaging for classification. In Forty-second International Conference on Machine Learning.
>
> [3] Xu, C., Yu, Y., Ren, H., Wang, Z., & Zou, C. (2025). Aggregating Conformal Prediction Sets via {\alpha}-Allocation. arXiv preprint arXiv:2511.12065.
>
> [4] Qin, S., He, J., Kuang, Q., Gang, B., & Xia, Y. (2024). Data-light Uncertainty Set Merging with Admissibility. arXiv preprint arXiv:2410.12201.

---

> > ### Author Rebuttal · Reviewer_5dr6 · 2026-04-01
> >
> > I thank the authors for their response. My concern regarding the choice of benchmarks for conformal aggregation has been fully addressed.
> >
> > I have one remaining minor question that does not affect my overall positive assessment: why are the experiments replicated only 20 times? This seems less standard in the conformal inference literature and may introduce noticeable Monte Carlo error.
> > For instance, in Table 1 (K=10, RF), the reported coverage for ECCP is 0.887, which slightly deviates from the nominal finite-sample coverage. My impression is that this may be largely due to the limited number of replications. Since the authors explain that the proposed P2E calibration is not computationally burdensome, I would appreciate it if they could comment on the rationale for this choice.

---

> > > ### Author Response · Authors · 2026-04-02
> > >
> > > Thank you for your follow-up question. We agree that using more replications reduces Monte Carlo error in empirical coverage estimates.
> > >
> > > The choice of 20 replications is a pragmatic trade-off for the full experimental pipeline. While the P2E calibration itself is inexpensive, the experiments require repeated training and evaluation across datasets, models, methods, and values of $K$. To address your concern directly, we rerun the ECCP experiments with 100 random seeds.
> > >
> > >
> > >
> > >
> > >
> > > ### Table 1. ECCP coverage with 100 seeds — Boston $\alpha=0.1$
> > >
> > > | K | Model | ECCP Coverage (mean ± sd) |
> > > |---|---|---|
> > > | 5  | OLS   | 0.8992 ± 0.0298
> > > | 5  | RF    | 0.8928 ± 0.0323
> > > | 5  | Lasso | 0.8980 ± 0.0304
> > > | 10 | OLS   | 0.9002 ± 0.0281
> > > | 10 | RF    | 0.8938 ± 0.0317
> > > | 10 | Lasso | 0.9008 ± 0.0292
> > > | 20 | OLS   | 0.9007 ± 0.0303
> > > | 20 | RF    | 0.8914 ± 0.0349
> > > | 20 | Lasso | 0.8996 ± 0.0310
> > >
> > > ### Table 2. ECCP coverage with 100 seeds — Abalone $\alpha=0.1$
> > >
> > > | K | Model | ECCP Coverage (mean ± sd)
> > > |---|---|---|
> > > | 5  | OLS   | 0.8963 ± 0.0210 |
> > > | 5  | RF    | 0.8972 ± 0.0241 |
> > > | 5  | Lasso | 0.8975 ± 0.0222 |
> > > | 10 | OLS   | 0.8955 ± 0.0214 |
> > > | 10 | RF    | 0.8951 ± 0.0244 |
> > > | 10 | Lasso | 0.8980 ± 0.0212 |
> > > | 20 | OLS   | 0.8972 ± 0.0215 |
> > > | 20 | RF    | 0.8957 ± 0.0238 |
> > > | 20 | Lasso | 0.8969 ± 0.0213 |
> > >
> > > _Remark._ The coverage of Split-CP, conditional on the training and calibration data, follows a Beta distribution under mild conditions with variance approximately $\alpha(1-\alpha)/(n+2)$ [1]. This indicates why we can see the coverage deviate slightly from the target, especially for small $n$.
> > >
> > >  We hope we adressed your concern; and we will rerun the experiments with 100 seeds in the revised manuscript.
> > >
> > > ---
> > >
> > > [1] Tibshirani, Ryan. "Conformal prediction." UC Berkeley (2023).

---

### Official Review · Reviewer_d6hc · 2026-03-08

**Soundness:** 2
**Presentation:** 2
**Significance:** 2
**Originality:** 3
**Overall Recommendation:** 5
**Confidence:** 3

**Summary:**

This work attempts to find a tight p-to-e calibrator in the conformal prediction setting.

**Compliance With Llm Reviewing Policy:**

Affirmed.

**Final Justification:**

My concerns have been addressed and thus I changed my evaluation.

**Key Questions For Authors:**

1. This might have been discussed in prior works, but it would be helpful to elaborate on why we would want an e-value-based approach, e.g., when do we face situations where some form of "aggregation" is needed? Specifically, conformal prediction is a wrapper designed to work with arbitrary initial estimators. What do we expect to gain from aggregating multiple estimators, and is there only gain but no loss? Aren't e-value-based approaches generally more conservative?

2. The description in Section 1.1 seems a bit inaccurate.

First, $Y_{n+1} = y$ is an event rather than a "hypothesis," since it is random rather than a fixed statement.

More importantly, what the conformal p-value defined in (3) satisfies is the marginal condition

$$P(P_n(Y_{n+1}) \leq \delta) \leq \delta.$$

It does not provide the conditional condition

$$P(P_n(Y_{n+1}) \leq \delta \mid Y_{n+1} = y) = P(P_n(y) \leq \delta \mid Y_{n+1} = y) \leq \delta.$$

3. The proposed calibrator contains $C_{n,\alpha}$, whose existence is guaranteed by the intermediate value theorem, but whose exact value is not provided. It is unclear to me how a user can actually implement this method. I also cannot find how this was implemented in the experiments. It is somewhat surprising that this issue is not discussed anywhere in the paper, yet the method appears to somehow have been implemented.

4. In this regard, I do not agree with the statement that this work provides an explicit p-to-e calibrator, as the main theorem essentially provides "a large class of calibrators that contains a valid calibrator" rather than "a valid calibrator".

5. Consequently, at this stage, it is difficult to discuss the experimental results, as it is unclear exactly what method is being implemented.

6. Where is it proved that the calibrator (9) satisfy the conditions A and C?

7. What is the role of Proposition 2.4? It doesn't seem to be used in the construction (9) or the proof of the main Theorem?

8. Related to point 1 above, I also think the standard conformal prediction should be included as a baseline, in the experiments.

**Limitations:**

-

**Strengths And Weaknesses:**

I think the writing should be significantly improved---the overall flow does not feel very smooth.

My major concern is that, while the authors claim that they provide an explicit calibrator with nice properties, the main theorem only proves the existence of such a calibrator rather than actually providing one. Further details are provided below.

---

> ### Author Rebuttal · Authors · 2026-03-31
>
> Thank you for the review. In our view, the reject score seems stronger than the comments support. Several concerns appear to stem from misunderstandings: Theorem 2.6 provides the calibrator used in experiments, Proposition 2.4 establishes existence,  and computing $C_{n,\alpha}$ is straightforward. We clarify these points below.
>
> "*I think the writing should be significantly improved*"
>
> - We believe the paper is reasonably well structured. Section 1 introduces the background on Split-CP and e-values, which forms the basis for Section 2. Sections 4 and 5 then explain how this calibrator is used in our experiments. We are happy to improve the writing if you could indicate specific points we should adress.
>
> *Clarification about $C_{n,\alpha}$ implementation*
>
> - We thank the reviewer for pointing this out, we will add the following paragraph to clarify this point after the proof of Appendix B.3: "Since the existence of $C_{n,\alpha}$ is guaranteed, we compute it in practice in a straightforward manner. For fixed $n$ and $\alpha$, we solve a **one-dimensional root-finding problem**, $\Sigma(C) - 1 = 0$, using `scipy.optimize.root_scalar`. This computation is efficient and depends only on $n$ and $\alpha$. Once this constant is obtained, the P2E calibrator $F_{n,\alpha}$ is guaranteed to satisfy Conditions A–C."
>
>
>
> "*..I do not agree with the statement that this work provides an explicit p-to-e calibrator.._*"
>
>
> - We indeed provide an "explicit" calibrator in the sense that all its components are fully specified and computable from ($n,\alpha,s,C_{n,\alpha})$ to determine $F_{n,\alpha}$. The P2E calibrator is then used to calibrate the conformal p-variable into an e-variable, as explain in Section 4.
>
>
>
> "*..why we would want an e-value-based approach, e.g., when do we face situations where some form of "aggregation" is needed? ..What do we expect to gain from aggregating multiple estimators, and is there only gain but no loss? Aren't e-value-based approaches generally more conservative?_.. I also think the standard conformal prediction should be included as a baseline*"
>
> - The need for aggregation arises naturally in practical settings where multiple predictors are available (e.g., different models, folds, or calibration sizes), and selecting a Split-CP set from a single model may be suboptimal or can introduce selection bias [1].  The goal of aggregation is to combine their uncertainty in an aggregated prediction set satifying the coverage guarantee. Empirically, there can be some cases  where this can lead to tighter prediction sets than those of individual predictors.
> - That said, aggregation does involve a trade-off: poorly designed methods can indeed be conservative. **This is precisely where our contribution lies**. E-value-based approaches are not inherently conservative; *their behavior depends on how the e-values are constructed*. By using set-preserving P2E calibrators together with admissible e-value merging, our method avoids unnecessary conservativeness while maintaining valid coverage, as confirmed by our experiments. Appendix A further shows that alternative e-value constructions of the form [2] can be substantially more conservative.
>
>
> "*The description in Section 1.1 seems a bit inaccurate...*"
>
> - We confirm the last paragraph of Section 1 is **accurate** and use standard notations of the hypothesis-testing view of CP [3,4]. However, we recognize that this can lead to some confusion, as pointed out by the reviewer, that the null hypothesis $H_0$  does not refer to a conditional event. To avoid confusion, we will rewrite the probability as $P_{H_0}$ in the revised manuscript.
>
>
>
> "*Where is it proved that the calibrator (9) satisfy the conditions A and C?*"
>
> - Proposition 2.4 automatically establishes Conditions A and B, as we explain in equation (7). We thank the reviewer for pointing this out, as we did not prove condition C of smoothness and invertibility of $F_{n,\alpha}$, although it is straightforward from the sigmoid-based form of the calibrator.
>
> "*What is the role of Proposition 2.4?*"
>
> - Proposition 2.4 establishes the existence of our proposed class of calibrators, whereas Theorem 2.6 provides the concrete P2E calibrator used in our experiments.
>
> ---
>
> We hope the clarifications above address the issues you raised and encourage you to consider raising the score.
>
> ---
>
> [1] Yang, Y., & Kuchibhotla, A. K. (2025). Selection and aggregation of conformal prediction sets. *JASA*, 120(549), 435-447.
>
> [2] Balinsky, A. A. and Balinsky, A. D. Enhancing conformal prediction using e-test statistics. arXiv:2403.19082, 2024.
>
> [3] Vovk, V., Gammerman, A., and Shafer, G. Algorithmic Learning in a Random World. Springer, 2005.
>
> [4] Papadopoulos, H., Proedrou, K., Vovk, V., and Gammerman, A. Inductive confidence machines for regression. In ECML, pp. 345-356. Springer, 2002.

---

> > ### Author Rebuttal · Reviewer_d6hc · 2026-04-01
> >
> > My concerns have been addressed.

---

### Official Review · Reviewer_L93m · 2026-03-12

**Soundness:** 3
**Presentation:** 3
**Significance:** 2
**Originality:** 3
**Overall Recommendation:** 3
**Confidence:** 4

**Summary:**

This paper studies the relationship between p-value-based and e-value-based formulations of conformal prediction (CP). It argues that existing p-to-e calibration methods may fail to preserve the conformal prediction set, which can lead to a loss of efficiency. To address this, the paper proposes a set-preserving calibration approach that maps conformal p-values to e-values while preserving the original prediction set. The paper develops theoretical results characterizing this calibration and its statistical properties, and it discusses implications for efficiency and inference. The goal is to provide a principled bridge between conformal p-values and e-values without altering the resulting conformal set.

**Compliance With Llm Reviewing Policy:**

Affirmed.

**Final Justification:**

The rebuttal clarified several points and improved my view of the paper, although some concerns remain. Overall, I maintain my current assessment.

**Key Questions For Authors:**

- Q1: Does the paper include sufficiently convincing empirical or real-data evidence to demonstrate the practical value of the proposed calibration approach?

- Q2: Since the conformal prediction procedure in (2) already enjoys a coverage guarantee, can the authors explain more clearly why introducing conformal p-values and e-values is necessary, and what concrete advantage this provides in practice?

**Limitations:**

No. The paper should discuss its limitations more explicitly, particularly the practical necessity of the p-value/e-value reformulation and the lack of stronger empirical evidence demonstrating its usefulness.

**Strengths And Weaknesses:**

While the paper develops an interesting theoretical connection between conformal p-values and e-values, the practical motivation remains somewhat unclear, especially because the original conformal prediction procedure in (2) already has a valid coverage guarantee. The paper would benefit from a clearer explanation of why the p-value/e-value reformulation is needed and from stronger empirical evidence, ideally including real-data analysis.

---

> ### Author Rebuttal · Authors · 2026-03-31
>
> Thank you for your review. Please find below our response to your comments.
>
> "*Does the paper include sufficiently convincing empirical or real-data evidence to demonstrate the practical value of the proposed calibration approach?*"
>
>
> - For CCP, our results show that the proposed e-value-based methods are **consistently among the most efficient**, as shown in Fig. 2, while satisfying the desired $1-\alpha$ coverage guarantee, unlike standard p-value-based CCP baselines [1,2]. Our ECCP can also provide greater robustness when the underlying predictor is unstable, as discussed in Appendix C.
>
> - For CA, our methods are again the most efficient among the aggregation approaches we consider (for e.g the p-value merging method $P_{Agg}$ ). Tables 2 and 3 suggest that the gains come from **combining our P2E calibrator with admissible e-value merging, with further improvement from randomization**.
>
>
> "*Since the conformal prediction procedure in (2) already enjoys a coverage guarantee, can the authors explain more clearly why introducing conformal p-values and e-values is necessary, and what concrete advantage this provides in practice?*"
>
> - We agree that the Split-CP in (2) provides a valid coverage guarantee, however, our work addresses different settings: CCP and CA, which share similar ideas with the Split-CP procedure (like splitting the data, constructing a p-value to build the prediction set). In these settings, the question is  **how to combine information from multiple models or folds while maintaining validity and improving efficiency**.
> - Introducing e-values enables principled aggregation through admissible merging rules, which is not naturally supported by standard p-value-based approaches. Empirically, this leads to tighter prediction sets while preserving coverage. More broadly, e-values provide a flexible framework that extends CP to more realistic scenarios where standard methods are less adaptable.
>
>
> We hope this clarifies the main message of our work and addresses your concerns and encourage you to raise your score.
>
>
> ---
>
> [1] Vovk, V. Cross-conformal predictors. Annals of Mathematics and Artificial Intelligence, 74(1):9–28, 2015.
>
> [2] Gasparin, M. and Ramdas, A. Improving the statistical efficiency of cross-conformal prediction. In Forty-second International Conference on Machine Learning, 2025.

---

> > ### Author Rebuttal · Reviewer_L93m · 2026-04-03
> >
> > I appreciate the authors’ efforts in preparing the rebuttal and clarifying several points.

---

### Official Review · Reviewer_a9Tr · 2026-03-12

**Soundness:** 4
**Presentation:** 3
**Significance:** 4
**Originality:** 3
**Overall Recommendation:** 5
**Confidence:** 2

**Summary:**

Conformal prediction (CP) procedures typically use p-values to construct prediction sets. P-values limit applicability of the procedure when combining evidence from different data splits or aggregating evidence from different models. Prediction sets based on e-values allow natural aggregation of evidence in these settings, but the existing e-value formulations are not set-preserving, leading to substantial decrease of efficiency of the prediction sets.

This article proposes a new p-to-e calibrator that is set preserving, yielding substantial efficiency gains compared to existing p-to-e methods, while retaining all advantages of e-CP compared to CP based on p-values. The new method is justified theoretically and demonstrated in two applications.

**Compliance With Llm Reviewing Policy:**

Affirmed.

**Key Questions For Authors:**

1. In the first case study, it seems as though ECCP is generally superior over ECCP-ex and UR-ECCP-ex. Are there situations where you would regardless prefer those two methods over ECCP?

2. Relatedly, ECCP-ex and UR-ECCP-ex show much larger variance than ECCP and ECCP(AoN), and the difference between ECCP and ECCP(AoN) seems rather miniscule. Can you demonstrate (with a clearer visualization or perhaps an alternative experiment) your theoretical expectation that ECCP(AoN) produces larger sets than ECCP, and how quickly does ECCP(AoN) converge ECCP in performance?

**Limitations:**

Yes.

**Strengths And Weaknesses:**

The article is theoretically sound and well justified, and provides a method that may become highly relevant both for theoretical advances in the field of e-conformal prediction (and conformal prediction in general) as well as in practical applications.

Results comparing the developed ECCP methods are convincing only for the p-to-e calibrators that were shown to be set-inflating (F1 and F2). Compared to the AoN calibrator, the comparison is not exactly flattering the new methods. (Nevertheless, there are still practical reasons why the new ECCP method may be prefered over AoN ECCP).

There are slight issues with presentation; e.g., a stub sentence on line 111, Figure 2 could show a horizontal reference line at $1-2\alpha$ for the rather methods, rather than $1-\alpha$ for all methods.

---

> ### Author Rebuttal · Authors · 2026-03-31
>
> We thank you for the positive review on our paper. Below we provide a response to your main comments and questions.
>
> "*In the first case study, it seems as though ECCP is generally superior over ECCP-ex and UR-ECCP-ex. Are there situations where you would regardless prefer those two methods over ECCP?*"
>
> - Indeed, ECCP is the best method overall from the experiments of Section 5.1. The Ex-ECCP variants were designed as attempts to leverage the exchangeable and randomized exchangeable Markov inequalities that our e-value framework allows. However, they exhibit much larger variance. For this reason, we do not necessarily recommend these variants in practice. Nevertheless, we believe it is still valuable to report how they behave empirically, as this helps clarify both their strengths and their limitations.
>
> - Our ECCP method shows small variance compared to Ex-ECCP & UR-Ex-ECCP. This high variance was also observed by Gasparin et al. [1], suggesting that this may be due to the **asymmetric nature of the corresponding aggregation rule** (eq. (16)). In contrast, ECCP invovles symmetrically averaging $K$ e-values (eq. (15)).
>
> "*..the difference between ECCP and ECCP(AoN) seems rather miniscule. Can you demonstrate (with a clearer visualization or perhaps an alternative experiment) your theoretical expectation that ECCP(AoN) produces larger sets than ECCP, and how quickly does ECCP(AoN) converge ECCP in performance?*"
>
> - Empirically, the gap between ECCP and ECCP(AoN) becomes more visible as $n$ decreases (i.e., as $K$ increases). In the box plots of Fig.2, the Ex-ECCP variants enlarge the overall scale of the figure, which can make these differences harder to see. For clarity, we therefore report the ECCP results below in Tables 1 and 2.
>
> **Table 1. Boston**: mean ± standard deviation for prediction set length for selected ECCP variants across different $K$
>
> | K | Model | Metric | ECCP | ECCP(AoN) | ECCP($F_2$) | ECCP($F_1$) | ECCP($F_3$) |
> |-|-|-|-|-|-|-|-|
> |5|OLS|Length|14.475 ± 0.594| 14.517 ± 0.596 | 39.335 ± 2.508 | 64.308 ± 2.580 | 59.541 ± 2.754
> |5|RF|Length|9.609 ± 0.662|9.644 ± 0.669 | 34.339 ± 3.136 | 62.227 ± 2.778 | 57.442 ± 2.954
> |5|Lasso|Length|14.516 ± 0.611| 14.559 ± 0.611 | 39.435 ± 2.494 | 64.345 ± 2.578 | 59.548 ± 2.755
> |10|OLS|Length| 14.770 ± 0.650| 14.950 ± 0.670 | 57.208 ± 2.695 | 69.543 ± 2.945 | 61.401 ± 2.640
> |10|RF|Length|9.715 ± 0.487|9.850 ± 0.504 | 54.129 ± 2.936 | 67.946 ± 3.107 | 59.358 ± 2.821
> |10|Lasso|Length| 14.775 ± 0.642|14.947 ± 0.664|57.256 ± 2.689 | 69.558 ± 2.942 | 61.404 ± 2.642
> |20|OLS|Length|15.971 ± 0.817|16.573 ± 0.966|71.075 ± 2.753 | 74.456 ± 3.412 | 65.167 ± 3.185
> |20|RF|Length|10.632 ± 0.758|11.162 ± 0.949|69.419 ± 2.971 | 73.273 ± 3.613 | 63.286 ± 3.404
> |20|Lasso|Length|15.980 ± 0.849|16.586 ± 0.992| 71.086 ± 2.758 | 74.464 ± 3.416 | 65.170 ± 3.184
>
> **Table 2. Abalone dataset**:
>
> | K|Model|Metric|ECCP| ECCP(AoN) | ECCP($F_2$) | ECCP($F_1$) | ECCP($F_3$) |
> |-|-|-|-|-|-|-|-|
> |5|OLS|Length|6.637 ± 0.048| 6.638 ± 0.048 | 10.303 ± 0.254 | 26.958 ± 1.693 | 32.274 ± 1.684
> |5|RF|Length|6.685 ± 0.088|6.686 ± 0.088|10.225 ± 0.312 | 26.811 ± 1.698 | 32.239 ± 1.687
> |5|Lasso|Length| 6.534 ± 0.062| 6.535 ± 0.063| 10.598 ± 0.254 | 27.064 ± 1.667 | 32.332 ± 1.684
> |15|OLS|Length| 6.716 ± 0.085| 6.760 ± 0.083| 10.471 ± 0.278 | 30.978 ± 1.629 | 32.547 ± 1.627
> |15|RF|Length|6.760 ± 0.089|6.801 ± 0.086|10.361 ± 0.298 | 30.895 ± 1.631 | 32.501 ± 1.626
> |15|Lasso|Length| 6.661 ± 0.085| 6.705 ± 0.085|10.809 ± 0.279 | 31.115 ± 1.622 | 32.603 ± 1.619
> |20|OLS|Length|6.714 ± 0.081|6.723 ± 0.083|10.576 ± 0.282 | 32.019 ± 1.573 | 32.681 ± 1.638
> |20|RF|Length|6.745 ± 0.077|6.754 ± 0.077|10.506 ± 0.335 | 31.944 ± 1.590 | 32.634 ± 1.639
> |20|Lasso|Length|6.666 ± 0.088|6.676 ± 0.088| 10.929 ± 0.294 | 32.134 ± 1.580|32.734 ± 1.637
>
> - As expected, ECCP and ECCP(AoN) perform similarly for large $n$, but the gap widens as $K$ grows, matching the theoretical study. Additionally, our P2E is smooth and avoids collapsing evidence to zero compared to AoN.
>
>
>
> - It seems difficult to obtain a bound on the set-size difference between ECCP(AoN) and ECCP sets, since this depends on the aggregation of $K$ e-values as well as on the constant $C_{n,\alpha}$. Still, Figure 1 gives some intuition about how quickly $F_{n,\alpha}$  approaches $F_{AoN}$. More importantly, since **our carefully designed P2E calibrator dominates AoN** (see theorem 2.6), the aggregated prediction set produced by our method is always smaller, as we show in eq. (10), for all $n$ and $\alpha$ defined in proposition 2.4. In that sense, there is no practical need to use AoN instead.
>
>
>
> We thank you again and we hope this addresses your concerns.
>
> ---
>
> [1] Gasparin, M. and Ramdas, A. Improving the statistical efficiency of cross-conformal prediction. ICML 2025

---

> > ### Author Rebuttal · Reviewer_a9Tr · 2026-04-01
> >
> > My comments were addressed. Thank you.

---

### Decision · Program_Chairs · 2026-04-30

**Decision:**

Accept (regular)

**Comment:**

This work introduces a transformation (i.e. calibrator) that takes a valid conformal p-value as input and outputs a valid conformal e-value. The advantage of using conformal e-values compared to p-values is that it is straightforward to aggregate multiple valid e-values into a new valid e-value. The proposed calibrator allows more efficient aggregation than the basic All-or-Nothing calibrator, where efficiency corresponds to small conformal set sizes. This is made possible by allowing the calibrator to depend on the sample size. Reviewers generally appreciate the novelty, significance, and clarity of the submission. I recommend acceptance.